# SkyLadder: Better and Faster Pretraining via Context Window Scheduling

**Tongyao Zhu**[1,2]  **Qian Liu**[2*]  **Haonan Wang**[1]  **Shiqi Chen**[3]
**Xiangming Gu**[1]  **Tianyu Pang**[2]  **Min-Yen Kan**[1]
[1]National University of Singapore   [2]Sea AI Lab   [3]City University of Hong Kong
tongyao.zhu@u.nus.edu; liuqian.sea@gmail.com

## Abstract

Recent advancements in LLM pretraining have featured ever-expanding context windows to process longer sequences. However, our controlled study reveals that models pretrained with shorter context windows consistently outperform their long-context counterparts under a fixed token budget. This finding motivates us to explore an optimal **context window scheduling** strategy to better balance long-context capability with pretraining efficiency. To this end, we propose Sky-Ladder, a simple yet effective approach that implements a short-to-long context window transition. SkyLadder preserves strong standard benchmark performance, while matching or exceeding baseline results on long-context tasks. Through extensive experiments, we pretrain 1B-parameter models (up to 32K context) and 3B-parameter models (8K context) on 100B tokens, demonstrating that SkyLadder yields consistent gains of up to 3.7% on common benchmarks, while achieving up to 22% faster training speeds compared to baselines[2].

## 1 Introduction

The evolution of language models has been marked by a consistent expansion in context window sizes (Figure 1 left). While early models like GPT [39] and BERT [8] were limited to context windows of 512 tokens, subsequent models have pushed significantly beyond these bounds. GPT-2 [40] doubled this capacity to 1024 tokens, and with Large Language Models (LLMs) exceeding 1B parameters, this trend has continued: Llama [51] has a 2048-token window, followed by Llama-2 [52] (4096 tokens), and Llama-3 [13] (8192 tokens). The need for models to handle longer sequences during inference has fueled the rush to expand the context window. As models pretrained with longer context windows reduce document truncation and preserve coherence [9], there is a widespread belief that such models should perform comparably to, or even surpass, their shorter-context counterparts.

We question the common belief that larger context windows do actually improve performance. Close inspection of previous work reveals that there has yet to be a fair experimental setup for comparing models across different context windows while adhering to a fixed token budget. Using tightly controlled experiments, we test how changing only the context window size during pretraining impacts their performance. As shown in Figure 1 (right), our results indicate that models pretrained using shorter contexts always outperform long-context models, when assessed by their average performance across popular benchmarks. In addition, we verify that the performance gap is not eliminated by using advanced document packing strategies [13, 9, 44].

To ensure the model can ultimately process long sequences, the model still needs to be exposed to long sequences. However, given the finding that shorter context windows enhance performance on

---

[*]Corresponding author.
[2]Project code is at https://github.com/sail-sg/SkyLadder

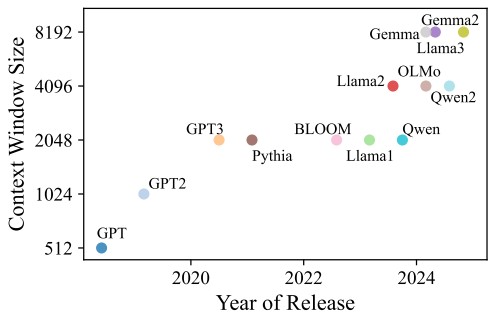 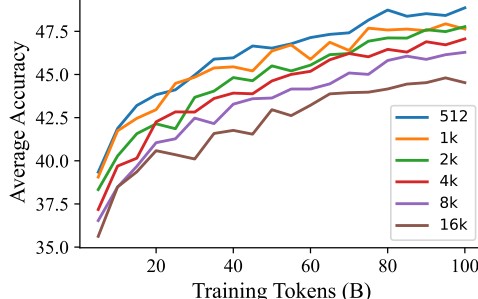

Figure 1: Left: Pretraining context window of LLMs grows over the recent years. Right: Average performance (in %) across nine downstream tasks for 1B-parameter models with different pretrained context window sizes (color-coded). Increasing the context window degrades the overall performance.

downstream tasks, we face a trade-off between long-context capability and pretraining effectiveness. We propose SkyLadder, a simple yet effective **context window scheduling** strategy designed to balance both objectives. SkyLadder does this by progressively expanding the size of the context window during pretraining, beginning pretraining with a minimal short context window (e.g., 8 tokens) and progressively expanding it to the long target context window (e.g., 32,768 tokens).

Empirical results on 1B-parameter models (up to 32K context window) and 3B-parameter models (up to 8K context window) on 100B tokens demonstrate that SkyLadder outperforms naive long-context pretraining baselines, in both short- and long-context evaluation tasks. For example, models trained with SkyLadder demonstrate significantly higher accuracy on standard benchmarks (e.g., HellaSwag), and reading comprehension tasks (e.g., HotpotQA), while still maintaining competitive performance on long-context evaluations like RULER. We further investigate the mechanisms behind the superior performance by observing the training dynamics, and discover that SkyLadder exhibits more concentrated and effective attention patterns.

Overall, we suggest that the length of the context window is an important dimension in pretraining and should be scheduled over the course of training. We recommend a progressive approach that begins with a small context of 8 tokens and gradually increases according to a linear function of training steps. Given a target context window (e.g., 32K), we suggest that allocating approximately 60% of the total training tokens to this expansion phase leads to stronger downstream performance compared to baselines. This scheduling strategy optimally enhances both training efficiency and model capability, offering a practical recipe for improving pretraining in language models.

## 2 Related Work

**Context Window Scheduling.** Early work explored gradually increasing the context window in smaller models like BERT and GPT-2, to improve training stability and efficiency [35, 28, 21]. Notably, Li et al. [28] proposed length warmup for more stable training but did not show clear performance gains, while Jin et al. [21] focused on training acceleration in 400M models. We extend these findings by demon-

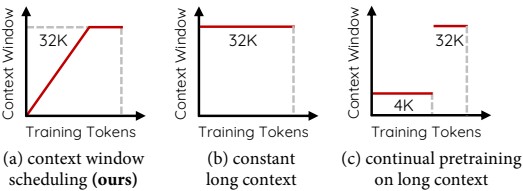

Figure 2: Schematic comparison of training-time context window scheduling.

strating, for the first time, that context window scheduling significantly boosts both *efficiency* and *performance* at much larger scales (up to 3B parameters). A parallel approach from Pouransari et al. [38] segments training documents by length, but Fu et al. [10] caution that such segmentation can introduce domain biases, as longer texts often cluster in specific domains such as books. Recent developments in continual pretraining with long context windows [37, 55, 12], can also be viewed through the lens of context window scheduling with different strategies (illustrated in Figure 2). Our work represents the first demonstration of both effectiveness and efficiency of context window scheduling, providing empirical evidence of its benefits in both standard and long-context benchmarks.

**Long-Context Language Models.** Long-context language models have received a lot of attention due to their ability to capture extended dependencies across large textual windows. Most existing

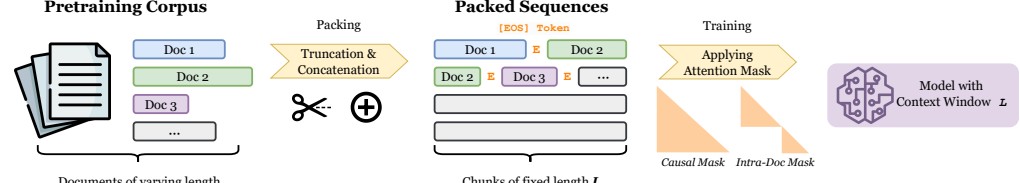

Figure 3: An illustration of the workflow for pretraining data preparation highlights several critical decisions. Key considerations include the method of data packing, the type of attention mask to employ (causal or intra-doc mask), and determining the appropriate context window length $L$.

approaches follow a *continual pretraining* paradigm [10, 57], which extends a pretrained backbone model to longer contexts through specialized fine-tuning or additional training. Several works propose to intervene in the positional embeddings to accommodate longer sequences [1, 31, 37, 4, 22], while others perform extended pretraining on longer-sequence corpora [12, 55, 32, 63]. Our approach differs from previous methods as we train *native* long-context models from scratch, rather than modifying a pretrained model in post-training. Compared with a naive long-context pretraining baseline with a constant schedule, our approach delivers substantial gains on multiple long-context tasks, underscoring the benefits of training from scratch. These findings show that our method can be a promising direction for future research on building language models with longer context windows.

## 3 How Context Window Affects Pretraining

*How does context window affect pretraining?* To investigate this in a fair and comparable manner, we pretrain language models from scratch with context windows ranging from 512 to 16,384 tokens under a fixed total number of tokens, evaluating via perplexity and downstream task benchmarks. We examine how the context window size impacts model performance, analyzing how data packing and masking strategies interact with window size.

### 3.1 Packing, Masking and Context Window

Most modern LLMs are based on a decoder-only transformer architecture [54] with a fixed context window size denoted by $L$. In contrast, the pretraining corpus, $D = \{d_1, d_2, d_3, \ldots, d_n\}$, consists of documents with varying lengths different from $L$. Therefore, a key step before pretraining is to *pack* the documents into sequences of length $L$. Formally, a packed sequence $C_i$ is constructed as $C_i = \text{Trunc}(d_{i,1}) \oplus d_{i,2} \oplus \cdots \oplus d_{i,n-1} \oplus \text{Trunc}(d_{i,n})$, where $\oplus$ represents concatenation, and $\text{Trunc}(\cdot)$ denotes truncation of documents to ensure $\text{len}(C_i) = L$. Following previous works [44, 64], document boundaries within $C_i$ are explicitly marked using end-of-sequence ([EOS]) tokens.

After the sequences are packed, the inputs are passed into transformer layers for next-token prediction training. A crucial component of these layers is the attention mechanism, which can be formulated as $A_{i,j} = q_i^\top k_j$, and then $\text{Attn}(X) = \text{Softmax}(A + M)$. In decoder-only models, a mask $M$ is applied to introduce constraints. A common approach is to use a *causal mask*, which ensures that each position can only attend to previous tokens by masking out (setting to $-\infty$) attention scores corresponding to future positions: $M_{ij} = -\infty$ for $j > i$ and $M_{ij} = 0$ otherwise. A recently proposed masking scheme, known as *intra-doc* mask [64, 13], imposes a constraint that only allows tokens to attend to each other if they belong to the same document. Let each document $d$ have start index $s_d$ and end index $e_d$, the masking can be denoted as $M_{ij}^{\text{intra}} = 0$ when $\exists d$ such that $s_d \leq i, j \leq e_d$ and $j \leq i$, and $M_{ij}^{\text{intra}} = -\infty$ otherwise. The model is trained with the standard cross-entropy loss on the packed sequences of length $L$. The workflow for pretraining data processing is illustrated in Figure 3.

### 3.2 Preliminary Study on Context Window Size

As per Section 1, we initiate our study by investigating the impact of context window size on model performance through a controlled experiment. Specifically, we pretrain language models with varying context window sizes, while preserving all other experimental settings. This enables a pure analysis of the context window's influence on model performance. Through this analysis, we aim to understand whether *longer context windows inherently lead to better or worse model performance.*

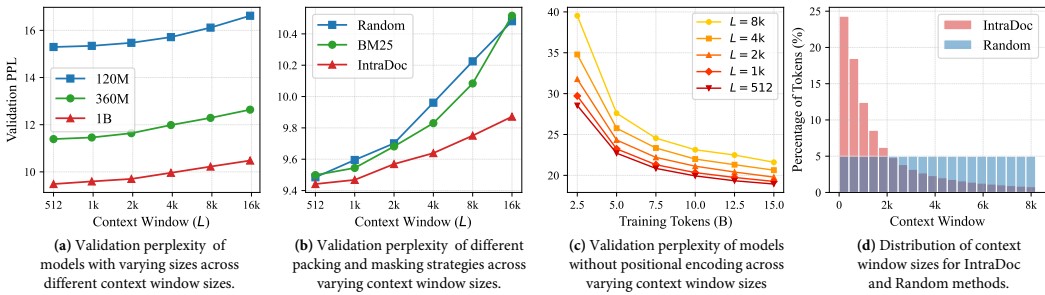

**(a)** Validation perplexity of models with varying sizes across different context window sizes.

**(b)** Validation perplexity of different packing and masking strategies across varying context window sizes.

**(c)** Validation perplexity of models without positional encoding across varying context window sizes

**(d)** Distribution of context window sizes for IntraDoc and Random methods.

Figure 4: Ablation studies of different factors on different context window sizes. Note that the validation PPL is obtained on the validation documents with a sliding window size of 512 tokens. The packing strategy in (a) is Random, and the model sizes in (b) and (c) are 1B and 120M, respectively. Note that the context window in (d) means the number of available preceding tokens when making next-token prediction (calculation details in Section A.6).

**Key Variables.** The context window size determines the number of tokens included in the context for each packed sequence. However, as discussed earlier, several additional factors influence the content within the context window: (1) *Packing methods* determine which documents constitute the context window, and different packing strategies can significantly alter the composition of token sequences; (2) *Masking methods* decide whether cross-document attention is enabled within the same context window. The choice of masking affects how the information from different documents interacts during training.

**Packing and Masking.** To study the impact of packing, we employ two strategies: *random packing* and *semantic packing*. For random packing, documents are randomly concatenated without a specific ordering. For semantic packing, inspired by Shi et al. [44], we retrieve and concatenate semantically relevant documents from the corpus, aiming to keep them within the same context window. After experimenting with both a dense retriever [20] and a lexical retriever BM25, we found that BM25 gives stronger performance and chose it as our focus. For masking, the baseline approach is causal masking, where each token can attend to all preceding tokens within the same context window, regardless of document boundaries. Conversely, recent studies [64, 9] show that disabling cross-document attention, thereby enabling intra-document attention, improves performance. For clarity in subsequent discussions, we denote random packing with causal masking as **Random**, BM25 packing with causal masking as **BM25**, and random packing with intra-document masking as **IntraDoc**.

**Training.** We pretrain models from scratch using the TinyLlama codebase [61], and study models with 120M, 360M and 1B parameters. Given the substantial computational cost associated with retrieval in semantic packing, we randomly select around 30B tokens from the CommonCrawl (CC) subset of the SlimPajama dataset [46] as the pretraining corpus. All models undergo training for up to 100B tokens (∼3.3 epochs). To ensure consistency across experiments, we strictly control all other settings, retaining the same batch size and learning rate schedule for all context windows. All models also incorporate Rotary Positional Encoding (RoPE) [47] to encode positional information. Appendix A.3 and A.4 give further model architecture details and training settings.

**Evaluation.** For all model sizes, we use perplexity (PPL) on validation documents from the original dataset as a key metric, in line with established practices [10, 24, 17]. Note that when comparing models across different context windows (e.g., a 2K-context model and an 8K-context model), we must ensure the evaluation sequence fits within the shorter model's context window to maintain a fair comparison. We also evaluate 1B models on downstream standard benchmarks: HellaSwag [60], ARC-Easy and ARC-Challenge [6], Winogrande [42], CommonsenseQA [48], OpenBookQA [34], PIQA [2], Social-QA [43], and MMLU [16]. We employ the OLMES suite [15] for the evaluation, as it has been shown to provide reliable and stable results with curated 5-shot demonstrations [12].

### 3.3 Experimental Results

Figure 1 presents the main experimental result, obtained using the Random setting with 1B-parameter models. The results indicate that context window size significantly influences the performance of LLMs, with *shorter contexts generally leading to better performance*. To further investigate the

factors contributing to the observation, we perform a comprehensive analysis to examine potential variables that may affect the conclusion. Figure 4 shows our results, and we derive four key findings:

> **Findings:** (1) The advantage of training on shorter contexts is consistent across model sizes; (2) This advantage is independent of the packing and masking methods employed; (3) It is also unrelated to the use of positional encoding; (4) The best packing and masking strategy is IntraDoc, which outperforms others probably because it introduces a larger number of short contexts during pretraining.

**Findings (1) and (2).** As shown in Figure 4, regardless of the model size in (a) or the packing and masking methods in (b), a shorter context window for pretraining generally results in higher average performance on benchmarks. The finding on benchmarks is consistent with the trend of validation PPL, where shorter context windows always yield lower PPL.

**Finding (3).** When using shorter context windows, one might hypothesize that the model learns positional encoding patterns for nearer positions more frequently, leading to better performance on standard benchmarks. To test the hypothesis, we systematically ablate RoPE by completely excluding it during pretraining, following prior work [25]. In Figure 4(c), models trained with short-context windows still outperform their long-context counterparts, even in the absence of positional encoding. This suggests that the advantages of shorter contexts are independent of positional encoding.

**Finding (4).** From Figure 4(b), we observe that IntraDoc achieves the best validation PPL across all context window sizes compared to Random and BM25, alongside consistently higher performance on standard benchmarks (c.f. Appendix A.7.1). This raises the question: why does IntraDoc excel? We attribute the advantage to the context window size distribution of IntraDoc, which implicitly increases the prevalence of shorter contexts. As illustrated in Figure 4(d), despite the sequence length of 8K, fewer than 1% of context windows actually reach this limit. While prior work links the success of IntraDoc to reduced contextual noise [64], we identify a complementary factor — reduced average context window size — as a key factor in its strong performance. That is, we hypothesize that the effectiveness of IntraDoc may also be closely tied to short context windows.

## 4 SkyLadder: Context Window Scheduling

We now present SkyLadder for progressively expanding the context window during pretraining.

### 4.1 Method

Inspired by learning rate scheduling, we explore whether dynamically scheduling the context window from short to long during pretraining could lead to performance improvements. This method can be implemented by applying multiple local "mini" causal masks to a long, packed sequence. We illustrate this masking strategy in Figure 5.

Formally, we define a local window length $w$. The associated mask $M_w$ is defined as follows: $M_{ij} = 0$ when $\lfloor \frac{i}{w} \rfloor w \leq j \leq i$, and $M_{ij} = -\infty$ otherwise, where $\lfloor \frac{i}{w} \rfloor w$ calculates the largest multiple of $w$ that is less than or equal to $i$, effectively defining a block-wise attention mask for the query token at position $i$. We linearly adjust the window size upwards by a constant

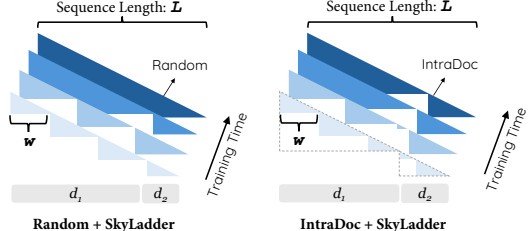

Figure 5: An illustration of SkyLadder with Random and IntraDoc. The example shows a packed sequence (length $L$) consisting of two documents. For SkyLadder, the context window $w$ starts from a small value and dynamically adjusts during training, eventually converging to the masking patterns of Random or IntraDoc.

factor per training step $t$: $w(t) = \min(w_e, w_s + \lfloor \alpha t \rfloor)$, where $w_e$ and $w_s$ represent the ending and starting context window sizes, respectively. Here, $\alpha$ denotes the rate of expansion, and $t$ corresponds to the training step. As the training progresses, when the dynamic context window size $w(t)$ eventually reaches the desired (long) context window size $L = w_e$, it remains fixed at that value. At this point, the attention mask is equivalent to a full causal mask. Notably, this method modifies

Table 1: Performance (accuracy in %) of different 1B models pretrained on 100B CC tokens on standard benchmarks. $^*$ denotes statistical improvements over the baseline (described in §A.7.3).

| Method | Avg. | ARC-E | ARC-C | CSQA | HS | OBQA | PIQA | SIQA | WG | MMLU |
|---|---|---|---|---|---|---|---|---|---|---|
| Random | 46.3 | 58.0 | 32.7 | 49.6 | 43.0 | 40.2 | 64.8 | 46.4 | 51.9 | 29.9 |
| + SkyLadder | **50.0** (+3.7) | **65.4**$^*$ | **35.6**$^*$ | **56.8**$^*$ | **47.0**$^*$ | **42.8** | 64.8 | **48.9**$^*$ | **56.0**$^*$ | **32.4**$^*$ |
| IntraDoc | 47.4 | 61.8 | 33.4 | 52.7 | 45.6 | 38.0 | 64.3 | 45.7 | 54.8 | 30.5 |
| + SkyLadder | 49.3 (+1.9) | 64.8$^*$ | 33.8 | 55.4$^*$ | **47.9**$^*$ | 39.4 | **66.1**$^*$ | 48.0$^*$ | 56.4 | 31.8$^*$ |

Table 2: Performance (accuracy in %) of 1B models pretrained on 100B CC tokens with different methods on reading comprehension and long-context benchmarks. Detailed setup is in Appendix A.7.3.

| Method | Reading Comprehension Benchmarks | | | | | | Long Benchmarks | | |
|---|---|---|---|---|---|---|---|---|---|
| | Avg. | HotpotQA | SQuAD | NQ | TriviaQA | RACE-h | Avg. | MDQA | RULER |
| Random | 25.5 | 6.5 | 37.0 | 15.8 | 37.7 | 30.7 | **15.3** | 17.7 | **12.8** |
| + SkyLadder | 30.2 (+4.7) | **12.4** | **40.2** | **20.4** | **43.0** | **35.0** | 14.3 | **18.3** | 10.3 |
| IntraDoc | 28.7 | 11.4 | 39.0 | 18.2 | 42.3 | 32.3 | 13.0 | 15.3 | 10.6 |
| + SkyLadder | 29.1 (+0.4) | 11.0 | 38.5 | **20.4** | 41.5 | 34.3 | 13.2 | 15.6 | 10.7 |

the effective context window through masking, independent of how the sequences are packed. As such, this mask $M_w$ can be integrated with $M^{\text{Intra}}$, which maintains the attention boundaries between documents; it can be seamlessly combined with most packing and masking strategies.

## 4.2 Experimental Setup

We follow the same setup in Section 3.2 to pretrain language models with 8K context on 100B tokens. We set $w_s = 32$ and $\alpha = 1/8$ by default, which means that a model roughly needs 64K steps (around 64B tokens) to reach the final desired context window of $L = 8192$. All baseline and SkyLadder models are implemented with Flash Attention 2 [7] (pseudocode in A.5). We fix all other hyperparameters, such as the learning rate schedule, batch size, etc., for fair comparison. Due to resource constraints, we do not perform extensive hyperparameter search to obtain the best combinations for $w(t)$, $\alpha$, and $w_s$. In our ablation study, we show that these hyperparameters have a negligible impact on performance, as long as they are within a reasonable range.

For evaluation, we use the same suite mentioned in Section 3.2 with standard benchmarks. To evaluate the performance of long-context question answering within an 8K length, we utilize the 30-document setting from the Multi-Document QA (MDQA) benchmark [30]. This is a widely-adopted benchmark that is shown to be reliable for models of 1B scale [38, 64], with an average length of approximately 6K tokens. We also select synthetic tasks within RULER [18], as defined by Yen et al. [59]. We choose the setup of the task that fills up the model's target context window $L$.

## 4.3 Experimental Results

Tables 1 and 2 present the main results, highlighting significant improvements achieved by SkyLadder across standard benchmarks, reading comprehension tasks and long-context benchmarks. For instance, compared to the Random baseline, integrating SkyLadder yields notable performance gains on standard tasks such as MMLU (+2.5%), ARC-E (+7.4%), and HellaSwag (+4%). This suggests that models with SkyLadder excel at learning common knowledge during pretraining. Additionally, our method further improves the performance of the strong baseline IntraDoc across many benchmarks. Meanwhile, for realistic long-context benchmarks like MDQA, SkyLadder matches or exceeds baseline performance. For RULER, the performance difference is likely because of fluctuation caused by its synthetic nature and small size [55]. More long-context evaluation can be found in Section A.7.3, confirming that SkyLadder is comparable with or better than baselines on long-context evaluation. In addition to Random and IntraDoc, we also verify that SkyLadder improves the performance of the BM25 model on both short and long tasks (Section A.7.4).

To address potential concerns that the benefits observed in short contexts may stem from the high level of noise in CC, we conduct additional experiments using the FineWeb-Pro dataset [65], a carefully curated high-quality dataset containing 100B tokens. As shown in Table 4, improved data quality indeed leads to substantial performance gains. However, our key findings remain consistent: IntraDoc

Table 3: Performance (in %) of 1B models pretrained on 100B Python code data. We follow the protocol of Huang et al. [19] to evaluate on HumanEval [3] and BigCodeBench [66]. $t$ is the sampling temperature. SkyLadder shows consistent improvement especially for 32K-context models.

| | | HumanEval | | | BigCodeBench | | |
| | | Greedy | Sampling ($t = 0.8$) | | Greedy | Sampling ($t = 0.8$) | |
| $L$ | Method | Pass@1 | Pass@10 | Pass@100 | Pass@1 | Pass@10 | Pass@20 |
|---|---|---|---|---|---|---|---|
| 32K | Random | 17.7 | 32.4 | 51.8 | 9.0 | 16.1 | 19.7 |
| | + SkyLadder | **21.3** | **37.7** | **59.8** | **9.4** | **20.6** | **24.3** |
| 8K | Random | 22.0 | 37.2 | 61.0 | 9.9 | 19.3 | 23.6 |
| | + SkyLadder | **23.2** | **38.2** | **63.4** | **11.3** | **20.0** | **24.1** |

Table 4: Performance (average accuracy over tasks, in %) of 1B models pretrained on FineWeb-Pro with an 8K context window.

| Method | Standard | Long |
|---|---|---|
| Random | 52.5 | 11.1 |
| + SkyLadder | **55.2** (+2.7) | 12.3 (+1.2) |
| IntraDoc | 54.3 | 12.7 |
| + SkyLadder | 54.8 (+0.5) | **13.9** (+1.2) |

Table 5: Performance (average accuracy in %) for models of different sizes.

| Size | Method | Standard | Long |
|---|---|---|---|
| 120M | Random | 40.1 | 5.8 |
| | + SkyLadder | 41.2 (+1.1) | 5.1 (-0.7) |
| 360M | Random | 47.2 | 8.9 |
| | + SkyLadder | 49.6 (+2.4) | 8.9 |
| 3B | Random | 57.0 | 15.8 |
| | + SkyLadder | **60.5** (+3.5) | **19.3** (+3.5) |

continues to outperform Random, and SkyLadder consistently delivers significant improvements over both baselines. This demonstrates that our method generalizes to corpora of varying quality.

We further examine whether SkyLadder is generalizable beyond natural language tasks. Following Ding et al. [9], we pretrain 1B code models on 100B Python code with the Starcoder tokenizer [29]. We observe a lower training loss ($\sim 0.9$) for code pretraining compared to natural language ($\sim 2.1$), suggesting that the structure in code makes the training easier. However, as shown in Table 3, there is still significant improvement when applying SkyLadder under both greedy decoding and sampling setups, especially when the target context length is 32K. This demonstrates the potential of SkyLadder to coding and possibly other reasoning tasks beyond natural language modelling.

### 4.4 Scalability Experiments

We examine whether SkyLadder's improvements persist as we scale up the model parameters and extend the context window size. We use the largest model and context size that our compute permits.

**Model Size.** We conduct experiments across three model sizes: 120M, 360M, and 3B parameters on the Fineweb-Pro dataset. Table 5 demonstrates that models utilizing SkyLadder consistently achieve better standard benchmark performance on all model sizes. For long context tasks, our method does not benefit 120M models, possibly due to their limited capacity in processing long sequences. However, the performance gain on 3B models is prominent. We observe a positive scaling trend: as the model size grows, the performance improvement also increases, indicating the potential of applying our method to even larger models beyond our current scale. We leave it as a future work to explore larger models as it requires significantly more compute.

**Context Window Size.** To examine whether SkyLadder can effectively scale to longer context windows, we train 1B models with a 32K context window on 100B FineWeb-Pro tokens. We adjust $\alpha$ to $1/2$ to ensure that the final context window expands to 32K before the end of pretraining. As shown in Table 6, our model demonstrates strong performance on both standard and long benchmarks. In addition, the performance difference of SkyLadder (0.9%) between the 8K and 32K models is largely reduced compared with the baseline approach (1.8%), which alleviates

Table 6: Performance (%) of 1B models trained on 100B FineWeb-Pro tokens with a 32K context window.

| Method | Standard | Long |
|---|---|---|
| Random | 50.7 | 9.7 |
| + SkyLadder | 54.3 (+3.6) | 13.5 (+3.8) |
| IntraDoc | 54.0 | 13.0 |
| + SkyLadder | **54.9** (+0.9) | **14.4** (+1.4) |

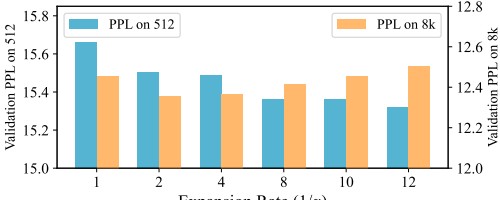 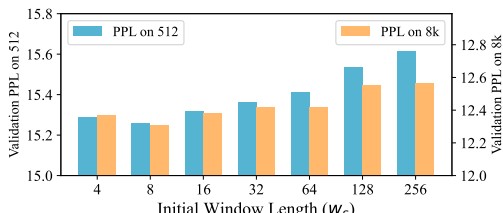

Figure 6: Validation PPL on 512 and 8K contexts of models with different expansion rate $\alpha$ (left) and initial window length $w_s$ (right).

Table 7: Comparison of 1B models trained with a 32K context window with different scheduling methods. Numbers are average accuracy (%).

| Method | Long | Standard |
|---|---|---|
| Constant Long (32K) | 9.7 | 50.7 |
| Linear (32→32K, *default*) | 13.5 | 54.3 |
| Stepwise Linear (32→32K) | 13.3 | **55.3** |
| Sinusoidal (32→32K) | **14.2** | 54.2 |
| Exponential (32→32K) | 11.5 | 54.7 |
| Cont. Pretrain (4K→32K) | 10.0 | 52.9 |

Table 8: Comparison of relative training time and compute efficiency for 1B Models with different context window sizes $L$. FLOPs calculation follows Zhang et al. [61]. A larger context window leads to more efficiency gains.

| Method | Time (%) | FLOPs ($10^{20}$) |
|---|---|---|
| Random (8K) | 100.0% | 11.6 |
| + SkyLadder | 86.9% (-13.1%) | 9.9 (-14.7%) |
| Random (32K) | 100.0% | 25.5 |
| + SkyLadder | 77.8% (-22.2%) | 18.8 (-26.3%) |

the performance degradation described in our earlier study. Notably, compared to the baseline Random approach, SkyLadder trains the model on progressively shorter contexts during earlier stages. This reveals a counterintuitive insight: naively training a model with a long context window is not always optimal, even if the model is evaluated on long contexts. In contrast, strategic scheduling of the context window during pretraining can yield better results.

## 4.5 Ablation Study

We now examine the impact of hyperparameters in SkyLadder scheduling. To manage computational costs, we adopt a default setup of pretraining 120M models with 8K context on 100B CC tokens.

**Expansion Rate.** We investigate the impact of the expansion rate $\alpha$ in Figure 6 (left). We choose different $\alpha$ ranging from slowest ($1/12$) to fastest ($1$). Our findings reveal that, for short contexts, performance generally improves as the expansion rate slows down. However, selecting an excessively slow rate (e.g., $1/12$) can negatively affect long-context performance due to insufficient training on longer contexts. **Therefore, we recommend setting $\alpha$ to $1/8$ for a good balance.**

**Initial Context Window.** As the final context window length $w_e$ is fixed to $L$, the sole remaining hyperparameter is $w_s$. Intuitively, setting $w_s$ to an excessively large value (e.g. close to $L$) leaves little room for scheduling, resulting in sub-optimal performance. In Figure 6 (right), we demonstrate that when $w_s$ is set to a relatively small value (e.g., 8), great performance can be achieved for both short and long contexts. This suggests that there is still potential for further improvement in our default setup. **Therefore, we recommend starting with a small context window, such as 8 tokens.**

**Scheduling Type.** The default scheduling method in SkyLadder is linear scheduling. We evaluate different context window scheduling types (more details in Table 20 and Figure 12 in Appendix A.7.4): (1) **Stepwise Linear** rounds window size $w(t)$ to multiples of 1K, resulting in a step function; (2) **Sinusoidal** increases quickly at the early stage then slows down; (3) **Exponential** starts slow but accelerates sharply; (4) **Continual pretraining** setup trains with 4K context windows for ~97B tokens, then switches to 32K context for the final 3B tokens. Table 7 shows that linear and sinusoidal schedules outperform the exponential variant on long tasks, likely because the exponential schedule, with extended short-context pretraining at the beginning, fails to adequately train on long contexts. Last, the most commonly used continual pretraining setup performs poorly overall, suggesting abrupt context changes harm both short and long performance. **These findings suggest that context window scheduling is superior to both constant long-context pretraining and continual pretraining.**

Overall, we conclude that the schedule should start from a small $w_s$ and the expansion should be gradual. We leave it to future work to study more advanced schedules and discover optimal configurations. For instance, it is possible that the schedule needs to be adjusted for various model sizes. More ablations for combination with BM25, hybrid attention, cyclic schedules and scheduling under a compute budget can be found in Appendix A.7.4.

## 4.6 Analysis and Discussion

**Training Efficiency.** We observe a significant boost in training efficiency when employing Sky-Ladder in Table 8. On 8K models, SkyLadder accelerates training time by 13% due to the reduced context window in calculating attention. With a 32K context window, the efficiency gain becomes even more pronounced: our method saves 22% of training time while achieving better performance. The FLOPs saving is larger than the actual time because of reduced attention calculation.

**Attention Pattern.** We next investigate why SkyLadder, despite being trained on short contexts overall, consistently outperforms the baseline. As language models rely on attention mechanisms to encode context information, we study how attention patterns change. Specifically, during pretraining, we monitor the dynamics of (i) attention entropy (solid lines in Figure 7), where a lower entropy is associated with better downstream performance [62]; (ii) attention sink [56], where the initial token in the context receives disproportionately high attention. We utilize the metric in Gu et al. [14] to quantitatively measure the amplitude of attention sink. As shown in Figure 7 (dashed lines), compared with the baseline Random, SkyLadder demonstrates reduced attention entropy, suggesting a more concentrated attention pattern. However, a slower emergence and lower amplitude of attention sink are simultaneously observed. This suggests that SkyLadder's attention is concentrated on the key information in the context instead of the initial token, which accounts for the performance gain.

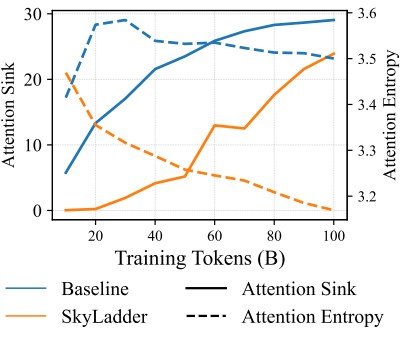

Figure 7: Dynamics of attention sink and entropy during pretraining 1B models (8K context). SkyLadder delays the emergence of attention sink while lowering the overall entropy, indicating a more effective attention pattern.

**Training Stability.** To further understand the reasons behind SkyLadder's better performance, we analyze the impact of pretraining context length on training dynamics. We pretrain 120M-parameter models with different context lengths. We first monitor the maximum attention logits ($S_{\max} = \max_{i,j} \ q_i \cdot k_j$ for all $i, j$) throughout pretraining, following the methodology of K2 [50]. A large attention logit indicates that an attention head is malfunctioning and may cause numerical instability. In Figure 8, we observe that pretraining with a long context of 16K tokens leads to exploding max attention logits, while a shorter window leads to lower attention logits.

Next, we study the loss and gradient behavior by computing four stability metrics over the first $N = 30K$ steps of pretraining, where $L_t$ denotes the training loss and $G_t$ is the gradient norm before clipping:

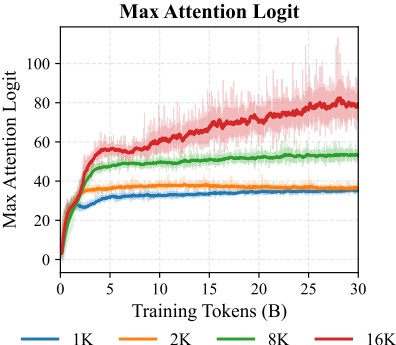

Figure 8: Max attention logits during training of models of different context lengths (in different colors).

- *Loss Volatility:* measures local fluctuations of loss over a sliding window ($w = 10$), computed as $\frac{1}{N} \sum_{t=1}^{N} \mathrm{Std}(L_{t-w+1}, \ldots, L_t)$. Lower values indicate more stable training.

- *Loss Smoothness:* the average loss change between consecutive steps, $\frac{1}{N-1} \sum_{t=2}^{N} |L_t - L_{t-1}|$. Smaller values mean smoother convergence.

- *Mean Loss Ratio* [28]: measures temporary increases in loss relative to the best loss so far, $\frac{1}{N-1} \sum_{t=2}^{N} \frac{L_t}{\min(L_1, \ldots, L_{t-1})}$, where smaller values indicate fewer loss spikes.

Table 9: Training stability metrics during pretraining of 120M models with different context lengths. All metrics are averaged over the first 30 billion tokens. ↓ indicates that smaller values are better.

| Context | Volatility ↓ ($w$=10) | Smoothness ↓ | Mean Loss Ratio ↓ | Avg Grad Norm ↓ |
|---------|----------------------|--------------|-------------------|-----------------|
| 1K | 0.023 | 0.019 | 1.014 | 0.335 |
| 2K | 0.026 | 0.023 | 1.017 | 0.338 |
| 4K | 0.030 | 0.029 | 1.020 | 0.340 |
| 8K | 0.036 | 0.036 | 1.025 | 0.347 |
| 16K | 0.041 | 0.042 | 1.036 | 0.416 |

- *Average Gradient Norm:* $\frac{1}{N} \sum_{t=1}^{N} \min(G_t, 1)$, where larger values indicate more aggressive gradient updates.

In Table 9, longer-context models show higher volatility, less smooth loss curves, more frequent upward spikes, and larger gradient norms, all indicating less stable optimization. In contrast, short-context models converge more smoothly with smaller fluctuations and more controlled gradient updates. Together, these results reveal that short-context pretraining is inherently more stable, both in attention behavior and optimization dynamics. The reduced numerical instability and smoother convergence likely enable more consistent gradient signals and better overall convergence, explaining their superior downstream performance.

**Comparison with Related Work.** We compare our method with another approach for improving pretraining in Table 10. As discussed in Section 2, Pouransari et al. [38] proposed Dataset Decomposition (DD) by segmenting a document into sequences of varying lengths and using a curriculum during pretraining. However, this approach inevitably introduces domain bias, as the document lengths in different domains are different [10]. This explains why DD with only one short-to-long cycle fails to outperform the IntraDoc baseline. To mitigate this, the authors suggested iterating through multiple cy-

Table 10: Comparison between SkyLadder and Dataset Decomposition (DD) on 1B models trained with 100B FineWeb-Pro tokens. Numbers are in average performance in %.

| Model | Standard | Long |
|-------|----------|------|
| IntraDoc | 54.3 | 12.7 |
| + SkyLadder | **54.8** (+0.5) | **13.9** (+1.2) |
| + DD (1 cycle) | 53.9 (-0.4) | 12.3 (-0.4) |
| + DD (8 cycles) | 54.5 (+0.2) | 13.5 (+0.8) |

cles of long and short data, which does improve performance substantially. In contrast, our method achieves better performance by avoiding such biases by not altering the data order based on length. In Appendix A.7.4, we experimented with various cyclic schedules but did not observe any improvements. In fact, we noticed loss spikes between cycles (Figure 14), indicating potential issues with domain shifts. This further supports that our method is safer since it does not disrupt the natural ordering and distribution of the data. More discussion with other related works [28, 21] is in Section A.8, where we demonstrate that our work provides novel insights that scheduling the context window over the entire training time improves both efficiency and performance.

## 5 Conclusion

We conduct a comprehensive controlled study of the impact of context window on pretraining, revealing that a shorter context window is more beneficial to the model's performance on standard benchmarks. This debunks the trend of pretraining with longer context windows. We therefore propose SkyLadder to schedule the context window from short to long during pretraining, which gives substantial improvement in downstream performance and computational efficiency. We conclude that context window scheduling is an important dimension for pretraining, and deserves more consideration. In the future, we plan to explore more dynamic and performant scheduling strategies that adapt according to model size or pretraining data distribution.

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

Table 11: Model configurations for pretrained language models.

| Model | Tinyllama 1B | Tinyllama 120M | Tinyllama 360M | Llama3.2 3B |
|---|---|---|---|---|
| **Vocab Size** | 32000 | 32000 | 32000 | 32000 |
| **Layers** | 22 | 12 | 18 | 28 |
| **Heads** | 32 | 12 | 16 | 24 |
| **Embedding Dim** | 2048 | 768 | 1024 | 3072 |
| **Intermediate Size** | 5632 | 2048 | 4096 | 8192 |
| **Normalization** | RMSNorm | RMSNorm | RMSNorm | RMSNorm |
| **Normalization $\epsilon$** | $1 \times 10^{-5}$ | $1 \times 10^{-5}$ | $1 \times 10^{-5}$ | $1 \times 10^{-5}$ |
| **Query Groups** | 4 | 1 | 16 | 8 |
| **Bias** | No | No | No | No |
| **RoPE $\theta$** | 10000 if $L = 8K$ 
 1000000 if $L = 32K$ | 10000 | 10000 | 100000 |

## A  Appendix

### A.1  Limitations

While we perform extensive experiments to study the impact of context window on pretraining and demonstrate the effectiveness of SkyLadder, we acknowledge that there are still limitations to be addressed. First, we conduct experiments up to a 3B-model scale and 32K context length, while the latest large language models are typically much larger and capable of processing longer contexts. However, pretraining a large model with a long context window requires prohibitive computational resources beyond our budget. Within our computational capabilities, we have tried to demonstrate the generalizability of SkyLadder across corpora, context window size, model size, and downstream tasks. Thus, we leave it as future work to apply SkyLadder to larger models. Second, we do not include a theoretical analysis to explain the effectiveness of SkyLadder as we mainly focus on empirical insights. We suggest that future work may investigate the relationship between the context window and the training compute to obtain the optimal context window schedule.

### A.2  Broader Impacts

The work aims to investigate the impact of choices on context windows in language model pretraining and proposes a way to speed up pretraining by scheduling context windows. On the positive side, this improves the efficiency of language model pretraining, making it more accessible and reducing the carbon footprint. Moreover, it enhances the performance of pretrained language models, which may result in better downstream performance in applications. There might be potential misuse of pretrained language models, which is beyond the scope of this work.

### A.3  Model Architecture

In Table 11, we list the architecture choices of the models trained, including the 120M, 360M, and 1B models based on the TinyLlama architecture [61]. The 3B model is based on Llama3.2 architecture [13].

### A.4  Training Configurations

We include details of the training configurations in Table 12. All models, irrespective of size or context window length, are trained on this same set of hyperparameters. For most of the hyperparameter values, we follow the TinyLlama [61] project, therefore, our results are highly reproducible.

### A.5  Implementation

We provide the pseudocode for implementing SkyLadder with Flash Attention 2 [7]. The only change is to apply local causal masking with size $w$, and combine them with the original document boundaries under the IntraDoc scenario. It can easily be integrated into any model before calculating attention. The rest of the training pipeline remains unchanged.

Table 12: Hyperparameters setup for pretraining the language models. All pretrained models follow the same structure.

| Parameter | Value |
|---|---|
| Optimizer | AdamW |
| AdamW-$\beta_1$ | 0.9 |
| AdamW-$\beta_2$ | 0.95 |
| Learning Rate Schedule | Cosine |
| Peak Learning Rate | 4e-4 |
| Minimum Learning Rate | 4e-5 |
| Warmup Steps | 2000 |
| Gradient Norm Clipping | 1 |
| Total Steps | 100,000 |
| Global Batch Size | 1,048,576 ($2^{20}$) tokens |
| Weight Decay | 0.1 |

---

**SkyLadder with Flash Attention 2**

```
# q, k, v: RoPE-encoded query, key, value tensors
# doc_boundaries: EOS token positions per document
# is_intradoc: intra-document attention flag
# training_step: current global step
# L: maximum context window length

# get current window size
w = min(L, get_current_mask_length(training_step))

# breakpoints every w tokens (and at document boundaries if using IntraDoc
    masking)
mask_boundaries = np.arange(w, L, w)
if is_intradoc:
    mask_boundaries = np.union1d(mask_boundaries, doc_boundaries)

# compute max segment length & cumulative lengths for flash attention
max_seqlen = get_max_seqlen(mask_boundaries, L)
cu_seqlens = get_cu_seqlens(mask_boundaries, L)

attn = flash_attn_varlen_func(
    q, k, v,
    cu_seqlens,
    max_seqlen,
    causal=True
)
```

## A.6 Definition of Per-token Context Window

In Figure 4(d), we show the context window distribution difference between IntraDoc and Random. To clarify, the context window size refers to the number of preceding tokens available in the context window when making the next token prediction. This is different from (a) and (b), where the context length $L$ is the model's pretrained context window.

Formally, consider a token at index $i$ and an attention mask matrix $M$, where an entry $M_{i,j} = 0$ indicates that token $i$ can attend to token $j$, and $-\infty$ otherwise. The context window size $C_i$ for the $i$-th token is defined as $C_i = \sum_{j=1}^{i} \mathbf{1}\{M_{i,j} = 0\}$, where $\mathbf{1}\{\cdot\}$ is the indicator function that returns 1 when $M_{i,j} = 0$ and 0 otherwise. In essence, $C_i$ is the number of tokens available as context for the $i$-th token, and the distribution of $C_i$ over all pretraining tokens is in Figure 4(d).

For Random, the causal mask is triangular: the $i$-th token has a context window size equal to $i$ (i.e., $C_1 = 1$, $C_2 = 2$, etc.). Thus, the distribution of $C_i$ is uniform. In contrast, IntraDoc effectively shortens the context length by limiting the cross-document attention.

## A.7 Additional Results

### A.7.1 Context Window Study

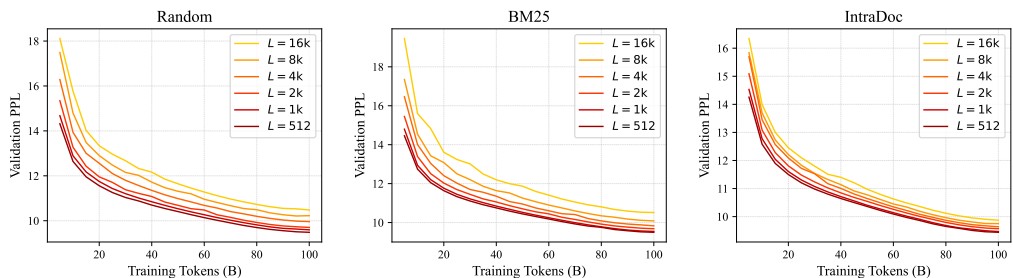

Figure 9: Validation perplexity (evaluated on a sliding window of 512) on models with different context lengths.

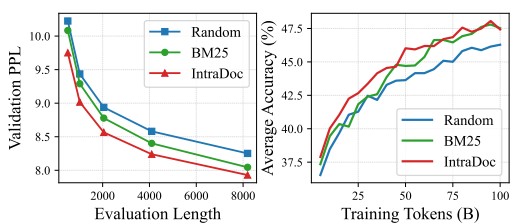

Figure 10: Left: Evaluation perplexity of models with different packing or masking strategies. Right: Downstream performance over 9 tasks of different models.

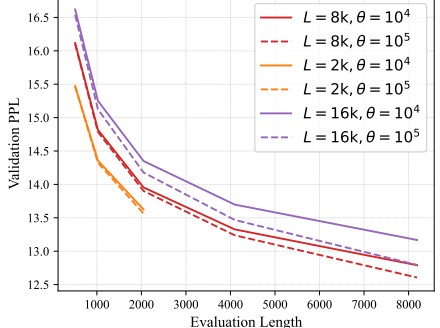

Figure 11: Validation perplexity vs training tokens with different context windows and base of RoPE, $\theta$. Evaluation is done on a sliding window of varying length (x-axis) on the validation documents.

In Figure 9. We plot the validation perplexity of models with different context windows under the Random, IntraDoc, and BM25 settings. We observe a consistent trend that a shorter-context model has lower evaluation perplexity on a shorter sequence under all settings.

In Figure 10, we plot the evaluation perplexity and downstream performance of models with different packing or masking strategies. We conclude that overall, IntraDoc achieves the best performance, with a consistently lower PPL and a higher downstream accuracy. We think that this is partially due to the shorter context window that the IntraDoc model is trained on.

### A.7.2 Ablations for Context Window Study

**Base of RoPE.** It has been shown that the value of RoPE may have a significant impact on the model's long context performance, and a longer context requires a larger base [33]. Therefore, we increase the RoPE base to 100,000, which is sufficiently large according to Men et al. [33]. In Figure 11, we observe an improvement for long-context models on long-context evaluation. However, the large gap between a shorter and a longer model still remains, therefore rejecting the hypothesis that the RoPE base is the key contributing factor to the superior performance of short-context models.

Table 13: Performance of 3B models on long tasks of retrieval-augmented generation (evaluated by exact-match scores) and reading comprehension benchmarks (accuracy in %).

| | Retrieval Augmented Generation | | | | Reading Comprehension | | |
|---|---|---|---|---|---|---|---|
| **Model** | Avg. | NQ | TriviaQA | HotpotQA | PopQA | Avg. | TOEFL | QuALITY |
| Random | 30.3 | 24.3 | 45.2 | 29.3 | 22.5 | 37.1 | 43.5 | 30.6 |
| + SkyLadder | **35.5** | **27.8** | **52.7** | **32.3** | **29.3** | **39.4** | **48.0** | **30.9** |

Table 14: Many-shot ICL performance (accuracy) on text classification benchmarks. Numbers in parentheses denote the number of labels for each task.

| Model | Avg. | DBpedia (14) | AGNews (4) | Amazon (2) | Yelp (2) | SST2 (2) |
|---|---|---|---|---|---|---|
| Random | 73.9 | 17.4 | 68.6 | **94.3** | 94.7 | **94.5** |
| + SkyLadder | **76.5** | **25.5** | **75.8** | 94.1 | **95.0** | 92.2 |

### A.7.3 SkyLadder Evaluation

**Statistical Test**  We test the statistical significance of the performance difference between our models and baselines in Table 1. We use a McNemar test as the two models are evaluated on the same set of questions. The original OLMES suite samples 1000 examples from each benchmark's full evaluation suite. In contrast, when conducting the McNemar test, we evaluate models on the full set to obtain more statistically meaningful results. We note that OpenBookQA only has 500 questions, making it harder to obtain statistical significance.

**Reading Comprehension**  For reading comprehension, we evaluate the following benchmarks: Hotpot QA (2-shot) [58], SQuAD (4-shot) [41], NaturalQuestions (NQ) (2-shot) [26], TriviaQA (2-shot) [23], and RACE-high (0-shot) [27]. We follow the setup by Zhao et al. [64], where NQ and TriviaQA use retrieved documents as contexts. For RACE, we use `lm-evaluation-harness` [11] to compare the PPL between options.

**Long-context Evaluation**  We provide additional long-context evaluation on our largest 3B model with an 8K context. This is to mitigate the performance instability of using synthetic benchmarks on small models. We first follow [38] to evaluate model accuracy on reading comprehension benchmarks TOEFL [5, 53] and QuALITY [36]. Next, we evaluate the model's performance on Retrieval Augmented Generation (RAG), where the model is provided with many relevant but potentially noisy contexts and needs to locate the correct information. As shown in Table 13, SkyLadder consistently performs better than the baseline across all evaluated RAG and reading comprehension datasets, highlighting its ability to locate correct answers within a lengthy context. In addition, we test the in-context learning ability of the models on text classification benchmarks [64, 44]. Results in Table 14 suggest that SkyLadder shows a significant gain for tasks with many labels, such as DBpedia, while achieving comparable high performance on binary tasks.

**Closed-book QA**  We additionally evaluate the closed-book QA performance of our models without access to any document. We use the evaluation protocol Zhao et al. [64] to measure the exact match. In Table 15, we notice a significant improvement in our methods compared to the baselines for answering closed-book questions. This is consistent with the results that our models show improvements on standard benchmarks that contain commonsense knowledge.

Table 15: 1B model (trained on CC) performance (exact match %) on closed-book QA tasks.

| | Closed-book QA | | |
|---|---|---|---|
| Model | NQ | TriviaQA | Average |
| Random | 6.1 | 11.9 | 9.0 |
| + SkyLadder | 9.0 | 17.5 | **13.2** |
| IntraDoc | 7.8 | 14.7 | 11.3 |
| + SkyLadder | 8.2 | 17.4 | **12.8** |

### A.7.4 SkyLadder Ablations

**Combination with BM25 Packing**  As SkyLadder only changes the context length via masking without altering the underlying data, it is orthogonal to any advanced data packing method such as Shi

Table 16: Performance (%) of 1B models with different schedule types. All models are trained on the same 100B CommonCrawl tokens with a final context length of 8K. BM25 packing, when combined with SkyLadder, significantly boosts performance on long tasks.

|  | Standard Avg. | Long Avg. |
|---|---|---|
| Random | 46.3 | 15.3 |
| BM25 | 47.5 (+1.2) | 16.4 (+1.1) |
| + SkyLadder | **49.8** (+3.5) | **17.0** (+1.7) |

Table 17: Performance (%) of 1B models with different schedule types. All models are trained on the same 100B FineWeb-Pro tokens with a final context length of 8K. Short-to-long scheduling is consistently better than long-to-short scheduling.

|  | Standard Avg. | Long Avg. |
|---|---|---|
| No Scheduling | 52.5 | 11.1 |
| Short-to-Long | **55.2** (+2.7) | **12.3** (+1.2) |
| Long-to-Short | 52.6 (+0.1) | 10.7 (-0.4) |

Table 18: Evaluation perplexity for Gemma3-like models with different evaluation context lengths $L_e$. All models are trained with 100B tokens on CommonCrawl.

| Model | $L_e = 512$ | $L_e = 4K$ | $L_e = 8K$ |
|---|---|---|---|
| Random – 120M | 15.9 | 13.4 | 13.0 |
| + SkyLadder | 15.5 | 12.9 | 12.4 |
| Random – 360M | 12.1 | 10.2 | 9.8 |
| + SkyLadder | **11.6** | **9.7** | **9.4** |

et al. [44], Ding et al. [9]. In Table 16, we combine the SkyLadder with the BM25 packing method. We show that the model achieves even better performance on both short and long context evaluation than BM25 without scheduling, which is also better than the Random baseline. This reveals that our method can be combined with more advanced packing techniques to further boost performance.

**Combination with Hybrid Attention** We note that a recent interesting trend in pretraining models with long context is to use a hybrid attention structure. For instance, Gemma3 [49] uses a mixture of global and local attention layers to balance efficiency and performance of the long-context model. We are curious about the generalizability of SkyLadder to such architecture, and follow Gemma3's strategy with a global-to-local ratio of 6:1. The results are presented in Table 18. We observe that SkyLadder consistently outperforms the baseline across all evaluation lengths, verifying its applicability. Importantly, SkyLadder works along the time dimension and is combinable with different attention variants, as long as there is a context window to be scheduled. We also verified the effectiveness of SkyLadder on alternative model structures. In Table 19, we pretrain models following the Qwen2.5-0.5B structure, and obtain consistent gains as well.

**Long-to-Short Schedule** A possibility that SkyLadder works better than baseline on standard benchmarks, which are typically short, might be that the training data mix has more short-context data after applying the mask. To study the effect of pure data distribution, we conduct an ablation of reversing the original short-to-long schedule and name it as the long-to-short schedule. This schedule spends the same number of tokens (64B) in the changing phase, before the constant training phase in $L = 8K$ for another 36B tokens. In Table 17, we show that the long-to-short schedule is not helpful to the model's performance in both short and long evaluation tasks. This highlights that the context window needs to be scheduled, rather than simply having a data mixture of long and short contexts.

**Alternative Schedule Types** We explore various types of short-to-long scheduling following different functions as mentioned in Section 4.5. Table 20 shows the details of the schedule as a function of $t$, and Figure 12 shows an illustration of the different schedule types. In Table 7, we show

Table 19: Evaluation perplexity for Qwen2.5-0.5B models under different evaluation context lengths $L_e$. All models are trained with 100B tokens on CommonCrawl.

| Model | $L_e = 1K$ | $L_e = 4K$ | $L_e = 8K$ |
|---|---|---|---|
| Random | 14.8 | 13.1 | 12.5 |
| + SkyLadder | 14.3 | 12.7 | 12.1 |

Table 20: Functions for different context window schedule types. We set $w_s = 32$ and $w_e = 32768$ in our experiments. The $r$ for rounding is set to 1024.

| Schedule | Function |
|---|---|
| Constant | $w_e$ |
| Linear | $w_s + (w_e - w_s)\frac{\alpha x}{w_e - w_s}$ |
| Stepwise | $\max(w_s, r \times \left\lfloor \frac{L(x)}{r} \right\rfloor)$ |
| Sinusoidal | $w_s + (w_e - w_s)\sin\left(\frac{\alpha \pi x}{2(w_e - w_s)}\right)$ |
| Exponential | $w_s \times \left(\frac{w_e}{w_s}\right)^{\frac{\alpha x}{w_e - w_s}}$ |

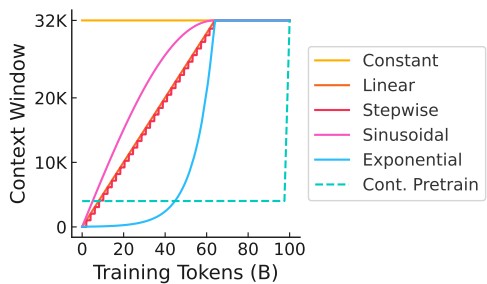

Figure 12: Illustration plot of various scheduling types.

that a smoother increase following the sinusoidal schedule works the best for long-context evaluation, while also achieving strong performance on standard benchmarks.

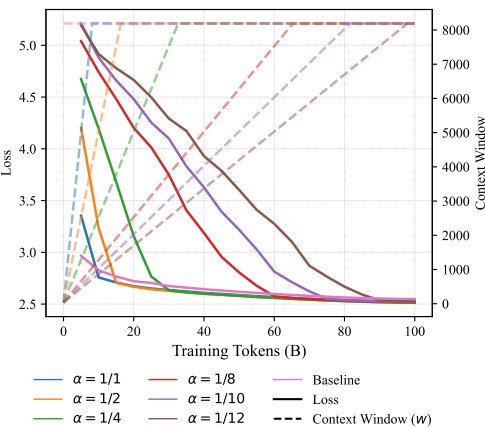

Figure 13: An illustration of the effect of different $\alpha$. Dashed lines represent the current context window $w$ for each step, and solid lines are the loss evaluated at 8K length.

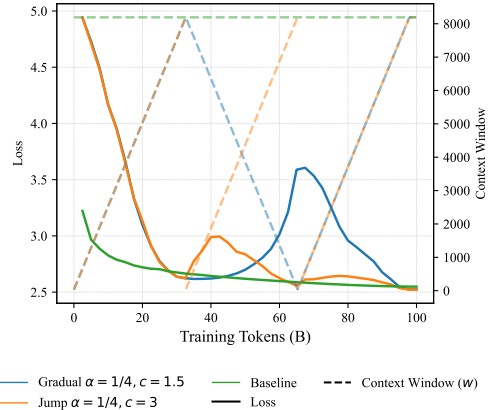

Figure 14: An illustration of the cyclic schedules with gradual increases or jumps. Dashed lines represent the context length for each step, and solid lines are the loss evaluated at an 8K length. $c$ represents the number of cycles.

**Expansion Rate**    We illustrate the effect of the rate of expansion $\alpha$ in Figure 13. As the evaluation is done on 8K contexts, models with a lower rate (and shorter context window) will have a higher loss as the evaluation length is out-of-distribution. However, eventually, all models' loss converges to a low level after the schedule reaches 8K. The detailed numbers of validation loss after pretraining can be found in Table 21. Following previous work [10, 17, 24], we consider a loss difference larger than 0.01 as significant. We conclude that setting a reasonable rate of $1/8$ balances both short and long-context loss, which is the default setup for our main experiments.

**Cyclic Schedule**    Inspired by the cyclic schedule learning rate [45], we also wonder if cycles are helpful in the schedule. In Figure 14, we show two cyclic schedules. In the "Jump" schedule, $w(t)$ will decrease to $w_s$ immediately after reaching $L$. On the other hand, the "Gradual" schedule means an "M" shape alternating between $w_e$ and $w_s$. Notably, in the discontinuous Jump schedule, we notice a significant increase in long-context perplexity when we train on only short contexts for an extended period. However, as long as $w$ increases back to $L$, the performance will return.

In Table 22, we show that these schedules have no major impact on the final performance. This highlights that the method does not introduce additional bias in data selection: different from existing methods such as Pouransari et al. [38] that proposes to train on short data first, followed by long data, we do not assume such a curriculum on data. We argue that the context window size should be independent of the data lengths to avoid bias in training only on certain domains of data.

Table 21: Validation loss with different expansion rates. A box is colored red if it is significantly worse (difference $> 0.01$) than the best of the column. $L_e$ is the evaluation context length. All models are of size 120M and trained on 100B tokens.

| Rate $(1/\alpha)$ | Tokens to Reach 8K (B) | $L_e = 512$ | $L_e = 4K$ | $L_e = 8K$ |
|---|---|---|---|---|
| 1 | 8 | 2.751 | 2.563 | 2.522 |
| 2 | 16 | 2.741 | 2.551 | **2.514** |
| 4 | 32 | 2.740 | **2.551** | 2.515 |
| 8 | 64 | 2.732 | 2.553 | 2.519 |
| 9 | 72 | 2.731 | 2.553 | 2.519 |
| 10 | 80 | 2.732 | 2.555 | 2.522 |
| 11 | 88 | 2.730 | 2.554 | 2.521 |
| 12 | 96 | **2.729** | 2.557 | 2.526 |
| Baseline (Constant) | | 2.780 | 2.590 | 2.549 |

Table 22: Validation loss with cyclic schedules. $L_e$ represents the evaluation context length. All models are of size 120M and trained on 100B tokens.

| Type | Number of Cycles | Tokens per Cycle (B) | $L_e = 512$ | $L_e = 8K$ |
|---|---|---|---|---|
| Random | | | 2.780 | 2.549 |
| + SkyLadder | | | 2.732 | **2.519** |
| Gradual | 4.5 | 16 | 2.743 | 2.530 |
| Jump | 9 | 8 | 2.744 | 2.532 |
| Gradual | 2.5 | 32 | 2.732 | 2.521 |
| Jump | 5 | 16 | 2.733 | 2.521 |
| Gradual | 1.5 | 64 | 2.728 | 2.524 |
| Jump | 3 | 32 | **2.727** | 2.522 |

**Initial Window Length**  We show the effect of having different $w_s$, the initial window length when the training starts. In Table 23, we show that the optimal starting length is 8 tokens. The trend is the same across both $\alpha = 1/4$ and $\alpha = 1/8$. This suggests that the starting length should be sufficiently small, irrespective of the expansion rate. It also reveals that prior studies, such as Jin et al. [21] and Pouransari et al. [38] that start with an initial length of 256 could be suboptimal.

**Compute Budget**  We show that when the total number of tokens is limited, our method can still improve language model performance. In Table 24, we choose 12.5B, 25B, and 50B total tokens as the computing budget, and vary the expansion rate so that $w$ reaches $L$ at the same point during training. We observe that under different token budgets, the performance trend is the best: gradually expanding the context window gives better performance than a rapid increase.

**Sliding Window Expansion**  A possible alternative to SkyLadder (using local causal masks by default) is to use a sliding window attention with a window size of $w(t)$ that changes with the training time. Formally, the mask becomes:

$$M_{i,j} = \begin{cases} 0 & \text{if } i - w \leq j \leq i \\ -\infty & \text{otherwise.} \end{cases}$$

so that each token in the context has a fixed preceding context of size $w$. When $w(t)$ reaches $L$, the mask becomes equivalent to a causal mask. We compare the performance of the two in Table 25 and observe that the sliding window approach shows slightly better performance in long tasks and worse performance in standard benchmarks. This is likely because overall there are more tokens with longer preceding contexts for the sliding window approach. In both cases, SkyLadder outperforms the Random baseline. We think that future work could further investigate the differences between SkyLadder implementations with causal and sliding window attention, such as the formation of attention sink [14]. There could also be possible combinations of the two: for instance, using a local mask first to disable distraction, and enabling sliding windows as the training progresses.

Table 23: Final validation loss after training 120M models on 100B tokens with different $w_s$ when $\alpha = 1/4$ and $\alpha = 1/8$. $L_e$ represents the context length of evaluation. A cell is colored red if its loss has a difference larger than 0.01 from the column's best. $w_s = 8192$ equals no scheduling.

| $w_s$ | $L_e = 512$ | $L_e = 4K$ | $L_e = 8K$ |
|---|---|---|---|
| | $\alpha = 1/4$ | | |
| 4 | 2.731 | 2.546 | 2.510 |
| 8 | **2.730** | **2.545** | **2.508** |
| 16 | 2.733 | 2.551 | 2.513 |
| 32 | 2.740 | 2.551 | 2.515 |
| 64 | 2.742 | 2.557 | 2.520 |
| 128 | 2.748 | 2.564 | 2.528 |
| 256 | 2.750 | 2.566 | 2.527 |
| | $\alpha = 1/8$ | | |
| 4 | 2.727 | 2.549 | 2.515 |
| 8 | **2.725** | **2.545** | **2.510** |
| 16 | 2.729 | 2.550 | 2.516 |
| 32 | 2.732 | 2.553 | 2.519 |
| 64 | 2.735 | 2.553 | 2.519 |
| 128 | 2.743 | 2.564 | 2.530 |
| 256 | 2.748 | 2.567 | 2.531 |
| 8192 | 2.780 | 2.590 | 2.549 |

Table 24: Final validation loss under different training token budgets and expansion rate $\alpha$ with 120M models. $L_e$ represents the context length used for evaluation. "% of Token Budget" means how many tokens are spent in the expansion phase with $w(t)$ increasing. Under all token budgets, we observe a consistent improvement when we spend around 64% in expansion, and 36% in the stable phase.

| $\alpha$ | Tokens to $L$ (B) | % of Token Budget | $L_e = 512$ | $L_e = 4096$ | $L_e = 8192$ |
|---|---|---|---|---|---|
| | | *Token Budget = 12.5B* | | | |
| 1 | 8 | 64% | **2.912** | **2.732** | **2.698** |
| 2 | 4 | 32% | 2.933 | 2.746 | 2.709 |
| 4 | 2 | 16% | 2.958 | 2.767 | 2.729 |
| 8 | 1 | 8% | 2.976 | 2.782 | 2.743 |
| | Baseline | | 3.008 | 2.823 | 2.790 |
| | | *Token Budget = 25B* | | | |
| 1/2 | 16 | 64% | **2.829** | **2.650** | **2.617** |
| 1 | 8 | 32% | 2.841 | 2.656 | 2.619 |
| 2 | 4 | 16% | 2.851 | 2.665 | 2.626 |
| 4 | 2 | 8% | 2.873 | 2.683 | 2.645 |
| | Baseline | | 2.918 | 2.734 | 2.700 |
| | | *Token Budget = 50B* | | | |
| 1/4 | 32 | 64% | **2.771** | **2.590** | **2.556** |
| 1/2 | 16 | 32% | 2.781 | 2.596 | 2.560 |
| 1 | 8 | 16% | 2.789 | 2.603 | 2.564 |
| 2 | 4 | 8% | 2.795 | 2.607 | 2.567 |
| | Baseline | | 2.839 | 2.652 | 2.616 |

## A.8    Additional Comparison with Related Work

We acknowledge that there are several prior works discovering a similar pattern of short-to-long pretraining. For instance, Li et al. [28] discover that using a sequence-length warmup for the initial steps in pretraining improves model stability. However, they mostly focus on stability in training loss and do not show a clear performance gain across multiple evaluations and larger scales. Moreover, we demonstrate that the benefits of scheduling a model's context window go beyond only the warmup stage. In Table 21's first row, simply warming up the model with 8B tokens results in suboptimal performance compared to a slower expansion rate. This validates that the context window should be

Table 25: Performance (%) of 1B models with different masking schemes. All models are trained on the same 100B FineWeb-Pro tokens with a final context length of 8K. Both implementations of SkyLadder outperform the baseline, and the sliding window approach excels at long tasks with a slight performance drop on standard benchmarks.

| Model | Standard Avg. | Long Avg. |
|---|---|---|
| Random | 52.5 | 11.1 |
| + SkyLadder w/ local causal | **55.2** (+2.7) | 12.3 (+1.2) |
| + SkyLadder w/ sliding window | 54.4 (+1.9) | **12.8** (+1.7) |

considered as a factor to schedule over the entire training course, which also differentiates us from Li et al. [28] that only consider the warmup stage.

Another related work is Jin et al. [21] where the authors use progressive sequence lengths to accelerate training. However, their method leads to worse performance under the same token budget, while our SkyLadder shows both time saving and performance improvement with the same number of tokens. We suspect that this might be because of the suboptimal schedule they used. Moreover, their study is limited to observing the training loss of small models (up to 410M parameters), while we comprehensively show performance gain across multiple corpora, model sizes, context sizes, and a wide variety of tasks. Overall, we systematically conduct controlled experiments on the impact of context window scheduling in pretraining, providing insights to explain these previous studies.

### A.9 Compute Information

We conducted all of our experiments for models with ≤ 1B size on an internal cluster of NVIDIA A100 nodes with 40G memory. Experiments with 3B models were conducted on H100 nodes. There are additional preliminary experiments that we did not include in the paper, which account for a fraction of the total compute. The detailed computation for each experiment is as follows: For the preliminary study on context window, pretraining a 1B model with 100B tokens (with 8K context) takes around 200 hours on a node of 8 A100s. Models of different sizes scale accordingly. For instance, plotting Figure 4(a) and (b) requires a total of 159 days of pretraining on a single node. For SkyLadder experiments, the baseline pretraining using various corpora takes the same time, and SkyLadder speeds up the training by 13% to 22% depending on the context length.

### A.10 Dataset Statistics

In this section, we provide detailed statistics of the datasets used in our study. These include the document length distributions of the pretraining corpora, the characteristics of the evaluation datasets, and the input length statistics of standard reasoning benchmarks.

Table 26 reports the document length statistics for the two pretraining corpora, *CommonCrawl* and *FineWeb-Pro*. Both distributions are strongly right-skewed, indicating that long documents are rare. Compared to FineWeb-Pro, CommonCrawl generally contains longer documents, while FineWeb-Pro has been more carefully cleaned and filtered.

Table 26: Document length statistics of the pretraining corpora, measured in tokens per document. Mean, median, and standard deviation describe the central tendency and variation. P25 and P75 indicate the 25th and 75th percentiles, while skewness and kurtosis capture distribution asymmetry and tail heaviness.

| Dataset | Mean | Median | StdDev | Min | Max | P25 | P75 | Skewness | Kurtosis |
|---|---|---|---|---|---|---|---|---|---|
| CommonCrawl | 1973 | 1067 | 4567 | 45 | 594,272 | 651 | 1867 | 21 | 820 |
| FineWeb-Pro | 1364 | 849 | 2295 | 1 | 230,949 | 507 | 1481 | 15 | 533 |

Table 27 shows the input length characteristics of common reasoning and knowledge benchmarks, including ARC, CSQA, HellaSwag, OBQA, PIQA, SocialIQA, Winogrande, and MMLU. While these benchmarks consist of relatively short contexts, they remain standard for assessing a model's factual consistency and reasoning ability. Importantly, a long-context model should maintain stable behavior even when the user provides a short query.

Table 27: Input length characteristics of common reasoning and knowledge benchmarks. Although relatively short, these tasks are crucial for measuring knowledge and reasoning consistency.

| Metric | ARC-C | ARC-E | CSQA | HellaSwag | OBQA | PIQA | SIQA | WinoG. | MMLU |
|---|---|---|---|---|---|---|---|---|---|
| Mean | 222 | 216 | 146 | 508 | 129 | 224 | 226 | 149 | 540 |
| Std | 20 | 16 | 7 | 32 | 9 | 29 | 6 | 4 | 512 |
| Min | 191 | 194 | 134 | 435 | 116 | 191 | 210 | 140 | 155 |
| Max | 401 | 336 | 203 | 576 | 192 | 440 | 262 | 170 | 3144 |

Finally, Table 28 summarizes the characteristics of the evaluation datasets used in the reading comprehension and long-context evaluation. These include QA benchmarks such as MDQA, RULER, SQuAD, HotpotQA, NQ, TriviaQA, and RACE. The datasets differ substantially in input length, reflecting the diversity of reasoning depth and context complexity.

Table 28: Length statistics of reading comprehension and QA evaluation datasets. These benchmarks capture varying levels of input complexity, from short factual QA to multi-hop reasoning tasks.

| Metric | MDQA | RULER | SQuAD | HotpotQA | NQ | TriviaQA | RACE |
|---|---|---|---|---|---|---|---|
| Mean | 5150 | 7259 | 1048 | 5010 | 583 | 566 | 492 |
| Std | 287 | 745 | 81 | 993 | 21 | 28 | 121 |
| Min | 4172 | 6209 | 923 | 3587 | 536 | 529 | 122 |
| Max | 6755 | 8061 | 1174 | 7842 | 633 | 643 | 1323 |

Overall, the datasets used in this work span a wide range of input lengths and domains, from large-scale pretraining corpora to short and long-context evaluation benchmarks, ensuring that our analysis is both comprehensive and representative.

## A.11 Licenses of Assets

We mainly use the following public datasets or codebases in this paper: SlimPajama [46] following the CommonCrawl Foundation Terms of Use[3], FineWeb-Pro [65] with an ODC-By 1.0 license, and TinyLlama [61] with an Apache 2.0 License.

---

[3]https://commoncrawl.org/terms-of-use

