# OpenReview forum: "SkyLadder: Better and Faster Pretraining via Context Window Scheduling"
_NeurIPS.cc/2025/Conference — NeurIPS 2025 poster_

### Official Review · Reviewer_hz6k · 2025-06-19

**Clarity:** 4
**Significance:** 3
**Originality:** 2
**Rating:** 4
**Confidence:** 4

**Summary:**

This paper uses tightly controlled experiments to demonstrate that the context length during large language model (LLM) pretraining can greatly affect the performance of the model on shorter-context tasks. More fine-grained experiments show that the document packing and masking strategies can also affect the performance. Based on these findings, the authors propose SkyLadder, a context length scheduler that dynamically adjusts the context length of the sequence during LLM pretraining by introducing "mini" causal masking that splits documents into several smaller segments. This scheduler improves the model performance while reducing the overall training time.

**Questions:**

1. The paper mentioned in Section 2 that for fair comparison, the context is limited to 512 tokens, which is the shortest context window of the tested candidate models. This seems to give an advantage to short-context windows as they were naturally trained with similar lengths. Given that RoPE, and especially NoPE can actually be extended to unseen lengths, have you ever tested the performance of these models on long-context tasks?

**Ethical Concerns:**

["NO or VERY MINOR ethics concerns only"]

**Final Justification:**

I will maintain my rating.

The long-context training scheduling isn't a new idea and has been explored by various research papers and technical reports. The findings mentioned in the rebuttal are not new. Here I will mention one more paper:

https://ai.meta.com/research/publications/effective-long-context-scaling-of-foundation-models/ (published in NAACL 2024)

Some of the ideas discussed by SkyLadder has already been discussed in this paper, e.g. longer isn't better, and context schedule is important. They have similar conclusions though in different settings.

The added experiments answered my question; thanks.

**Limitations:**

yes

**Quality:**

3

**Strengths And Weaknesses:**

## Strengths

1. The controlled experiments and the analysis in Section 3 are very useful and provide insights on the design of the pretraining process. Some conclusions, such as the effect of packing/masking strategies and the experiments without positional encoding, can shed light on future practices.
2. All the experiments, including those in Sections 3 and 4, are controlled with detailed explanations on reproduction. The conclusions are convincing.
3. This paper provides comprehensive analysis, including the effect of context length scheduling on attention patterns, scheduling type, and training efficiency.
4. The experiments are costly with multiple runs of the training processes. This systematic analysis done in this paper, as well as practices such as the expansion rate selection, can greatly save the efforts of future researchers.

## Weaknesses

1. Changing the context length is not a new idea. In past work, as early as Longformer, or many open LLMs with technical reports like DeepSeek v3, all mentioned that they would first pretrain the model on short contexts, and then gradually expand the context length till the maximum length. Though many of them did not systematically detail how the scheduling is done and the packing/masking solutions, they should not be too much different from SkyLadder.
2. Continuing from the above point, using a baseline that is done without scheduling can be misleading, as it is not the common practice of actual LLM pretraining. In Section 4.6 the authors discussed DD, while this is not the only method. It might be better to use a more appropriate method such as 2-stage training, i.e., short-context pretraining + long-context adaptation.

---

> ### Author Rebuttal · Authors · 2025-07-31
>
> Thanks for your valuable feedback and for recognizing the value of our controlled experiments. Here are our responses to your questions:
>
> ***Changing the context length is not a new idea.***
>
> We fully acknowledge prior works that employ the idea of growing the context length. However, as discussed in Section 2, these approaches were introduced primarily for efficiency reasons and lacked rigorous and public evaluation of their impact on final model performance. To our knowledge, no prior work has conducted a controlled, large-scale study across multiple model sizes, datasets, and masking strategies to quantify:
> 1. How pretraining context length affects downstream task performance under a fixed token budget.
> 2. How different scheduling strategies compare in terms of performance.
>
> For (1), our controlled experiments challenge the common assumption that longer context windows are better if compute permits. We show that naïve long-context pretraining does not consistently improve performance. This overturns the default “longer is better” expectation and highlights context length as a critical but underexplored dimension in pretraining design.
>
> For (2), while existing models often extend context in multiple stages, there has been no public and systematic analysis of how the schedule pace affects outcomes. In Table 7 (illustrated in Figure 11), we show that different schedules, despite reaching the same final 32K context, can lead to significantly different performance. This shows that a systematic investigation of the schedule is necessary and deserves more open and public research.
>
> ---
>
> ***It might be better to use a more appropriate method such as 2-stage training, i.e., short-context pretraining + long-context adaptation***
>
> We agree that current models often adopt a two-stage procedure: an extended period of training at a fixed mid-range length (e.g. 4K or 8K), followed by long-context continual pretraining on a much smaller token budget with gradual expansion. We therefore include a Cont. Pretrain baseline following this setup and compare it with SkyLadder.
>
> We experimented with pretraining 1B models with 32K context. The SkyLadder schedule starts from 32 at the beginning and gradually expands towards 32K. In contrast, the Cont. Pretrain setup spends 95% of training tokens at 4K length, before expansion to 32K with the rest 5% tokens. The performances are as follows:
>
> | Model                                 | Standard Avg. | Long Avg. | MDQA | RULER |
> |--------------------------------------|----------------|-----------|------|-------|
> | Constant 32K       | 50.7           | 9.7       | 12.8 | 6.6   |
> | Cont Pretrain (4K  $\rightarrow$ 8K  $\rightarrow$ 16K  $\rightarrow$ 32K) | 52.9           | 10.4      | 15.3 | 5.4   |
> | SkyLadder (32 $\rightarrow$ 32K)       | **54.3**       | **13.5**  | **18.4** | **8.5**   |
>
> While two-stage training improves over the constant baseline, SkyLadder achieves the highest performance on all tasks. This demonstrates that scheduling should be planned over the entire pretraining process, not only during the adaptation stage. This controlled comparison shows that SkyLadder provides genuine gains over the commonly used heuristic approaches.
>
> ---
> Here are our responses to the “Questions” Section:
>
> ***Given that RoPE, and especially NoPE can actually be extended to unseen lengths, have you ever tested the performance of these models on long-context tasks?***
>
> We evaluated whether short-context models can match long-context models when extended to 8K using RoPE adjustments or NoPE.
>
> First, we take RoPE-based models pretrained with 512–4K contexts and apply three common extension methods (Linear, Dynamic NTK, and YaRN) to scale the context window to 8K. The table below reports evaluation perplexity (PPL) at 8K. For reference, a model pretrained directly at 8K achieves a PPL of 8.25.
> | Pretraining Context Length | Linear   | DynamicNTK | YaRN   |
> |-----------------|----------|---------|--------|
> | 512             | 176.20   | 72.47   | 11.79  |
> | 1k              | 152.18   | 27.91   | 9.21   |
> | 2k              | 71.29    | 12.22   | 8.41   |
> | 4k              | 17.18    | 8.47    | 8.23   |
>
> **Observation**: YaRN provides the best results, but only the 4K→8K model approaches the 8K-pretrained baseline. Models trained on very short contexts (512 or 1K) perform poorly when extrapolated, showing that post-hoc scaling cannot fully substitute long-context pretraining.
>
> Next, we evaluate these extended models on the MDQA-30 document task. For reference, a model pretrained directly on 8K achieves a score of 17.7.
>
> | Pretraining Context Length | Linear | DynamicNTK | YaRN |
> |----------------------------|--------|------------|------|
> | 512                        | 0.4    | 0.8        | 5.9  |
> | 1k                         | 0.5    | 1.3        | 9.5  |
> | 2k                         | 1.0    | 4.5        | 15.4 |
> | 4k                         | 6.2    | 12.3       | 14.0 |
>
> **Observation:** Even with the strongest extrapolation techniques, short-context models underperform the 8K-pretrained model on MDQA-30. The performance gap widens as the pretraining context length decreases.
>
> For NoPE models, we encountered training stability issues and were able to do small-scale experiments (120M parameters, with 15B tokens). Due to the limited model size, we only evaluated perplexity (PPL) on held-out validation text, with $L_e$ indicating the evaluation context window.
>
> | Pretraining Context | $L_e=512$ | $L_e=1K$ | $L_e=2K$ | $L_e=4K$  | $L_e=8K$ |
> |-------------------------|---------|--------|--------|--------|--------|
> | 512                     | 18.9    | 37.4   | 213.9  | 767.9  | 1454.1 |
> | 1K                      | 19.3    | 17.9   | 61.0   | 447.0  | 2107.3 |
> | 2K                      | 19.8    | 18.3   | 17.2   | 61.6   | 1308.5 |
> | 4K                      | 20.6    | 19.0   | 17.8   | 16.9   | 37.6   |
> | 8K                      | 21.6    | 19.8   | 18.5   | 17.5   | 16.5   |
>
> **Observation**: PPL increases sharply whenever the evaluation length exceeds the pretraining length. While NoPE models show some extrapolation ability, they still fail to match the performance of models trained natively with longer contexts.
>
> Overall, both RoPE-based extensions and NoPE provide only limited long-context capability. **Short-context models cannot fully substitute dedicated long-context pretraining**, especially on complex long-document tasks. This highlights the necessity of long-context training. Our paper thus investigates the challenges in long-context pretraining and proposes SkyLadder to improve both long and short performance.
>
> We hope that our response has addressed your concerns, and we would greatly appreciate it if you could consider increasing the score.

---

### Official Review · Reviewer_78Fj · 2025-06-20

**Clarity:** 3
**Significance:** 3
**Originality:** 2
**Rating:** 4
**Confidence:** 3

**Summary:**

In this paper, the authors conduct a comprehensive controlled study on the impact of context window size during pretraining, revealing that shorter context windows lead to better performance on standard benchmarks. This finding challenges the prevailing trend of using longer context windows for pretraining. Based on this insight, the authors propose SkyLadder, a scheduling strategy that gradually increases the context window from short to long during pretraining. SkyLadder yields substantial improvements in both downstream performance and computational efficiency.

**Questions:**

As mentioned in the weaknesses, I would like to ask:

1. Could the authors provide a more in-depth analysis of why short-context pretraining leads to better performance?

2. Why does the proposed SkyLadder method fail to improve performance on certain reading comprehension tasks, as shown in Table 2?

**Ethical Concerns:**

["NO or VERY MINOR ethics concerns only"]

**Final Justification:**

Thanks to the authors for the thorough analysis, which solves my concerns on (1) the novelty of idea and (2) why the proposed method fails to improve performance on certain reading comprehension tasks. Therefore, I remain positive about this paper.

**Limitations:**

yes

**Quality:**

3

**Strengths And Weaknesses:**

**Strengths:**

1. Overall, I appreciate the writing quality of this paper. The plots are clear and well-organized, and the authors provide a solid background on long-context pretraining. All implementation details, ablation studies, and analyses are clearly presented and easy to follow.

2. The authors conduct extensive experiments across different datasets, model sizes, hyperparameters, and settings to support their motivation. The proposed method is simple yet effective.

3. The thorough experiments and analyses contribute valuable insights that could benefit future research on language model training.


**Weaknesses:**

1. While the paper is well-written and clear, the core idea is not particularly surprising. Moreover, the motivation behind the proposed method appears to be primarily result-driven, with limited theoretical or in-depth analysis. For example, it would be helpful if the authors could analyze *why* shorter contexts work better—possibly from the perspective of learning dynamics or optimization behavior.

2. In Table 2, the gains from applying SkyLadder are not entirely consistent—e.g., 3 out of 5 tasks show performance drops. This weakens the claim that SkyLadder improves reading comprehension. It would be helpful if the authors could elaborate more on these results and explain the underlying factors.

---

> ### Author Rebuttal · Authors · 2025-07-31
>
> Thanks for your helpful feedback. We appreciate that you found our work clear and valuable. Here are our responses to your questions:
>
> ***The core idea is not particularly surprising.***
>
> Our contribution is the first large-scale, controlled study across multiple model sizes, datasets, and masking strategies, demonstrating that shorter windows consistently outperform long ones under a fixed token budget. These findings challenge the current trend of pretraining with longer context windows (Figure 1).
>
> Previous works may apply short-to-long pretraining for efficiency reasons without showing a clearer performance gain, as discussed in Section 2. We show that proper context scheduling can improve performance on both short and long tasks, in addition to training speedup. We verified this effect across model sizes and pretraining corpora. This reveals that the schedule context length is a critical yet underexplored dimension of pretraining design.
>
> ---
>
> ***More analysis on why short context pretraining performs better.***
>
> Short-context pretraining provides more stable learning dynamics overall. We identify three key observations below:
>
> First, as increasing the context window mostly affects the attention layer, we choose it as the focus of our study. In Section 4.6 Figure 7, we plot the dynamics of attention sink and entropy during the pretraining: we observe that **SkyLadder delays the emergence of attention sink**. This is consistent with the findings of concurrent work [1] that short-context pretraining results in a lower attention sink. Overall, short-context models have more active attention heads that do not sink to the first token, leading to more capacity in processing the context.
>
> Second, we monitor the max attention logits $S_{\text{max}} = \max_{i,j} q_i \cdot k_j$ across the training steps, following a recent study of K2 [2]. A large attention logit suggests that a head is malfunctional and leads to numerical instability. We pretrained 120M models with different context lengths (each row), and get $S_{\text{max}}$ as training progresses (each column represents a checkpoint pretrained with X tokens).
>
> | Training Context | 2B   | 5B   | 10B  | 15B  | 20B  | 30B  | 40B  |
> |-------|------|------|------|------|------|------|------|
> | 1K    | 19.2 | 28.2 | 32.3 | 33.0 | 33.9 | 34.9 | 35.3 |
> | 2K    | 23.7 | 34.8 | 37.3 | 37.7 | 37.2 | 36.5 | 36.0 |
> | 8K    | 16.3 | 36.1 | 46.8 | 49.5 | 51.2 | 52.8 | 54.0 |
> | 16K   | 21.3 | 49.8 | 56.7 | 63.5 | 69.3 | 77.7 | 84.4 |
>
> We observe that the 16K model shows growing $S_{\text{max}}$, and still increases steadily at the end of training. In contrast, the 1K model exhibits smaller $S_{\text{max}}$ overall, and the value quickly saturates after 5B tokens. Thus, **short-context models give smaller max attention logits**.
>
> Finally, we directly monitor the training dynamics by investigating the training loss and gradient behavior. We compute four stability metrics for the first $N = 30K$ steps of pretraining, where $L_t$ is the training loss and $G_t$ is the gradient norm before clipping:
>
> - **Loss Volatility:** measures local fluctuations of loss over a sliding window ($w=10$), computed as $\frac{1}{N} \sum_{t=1}^{N} \text{Std}(L_{t-w+1}, \dots, L_t)$. Lower values indicate more stable training.
> - **Loss Smoothness:** the average loss change between consecutive steps, $\frac{1}{N-1} \sum_{t=2}^{N} |L_t - L_{t-1}|$. Smaller values mean smoother convergence.
> - **Mean Loss Ratio:** measures temporary increases in loss relative to the best loss so far, $\frac{1}{N-1} \sum_{t=2}^{N} \frac{L_t}{\min(L_1, \dots, L_{t-1})}$, where smaller values indicate fewer loss spikes.
> - **Average Gradient Norm:** $\frac{1}{N} \sum_{t=1}^{N} \min(G_t, 1)$, where larger values indicate more aggressive gradient updates.
>
> | Context | Volatility ↓ (w=10) | Smoothness ↓ | Mean Loss Ratio ↓ | Avg Grad Norm ↓ |
> |---------|--------------------|--------------|-------------------|-----------------|
> | 1K      | **0.023**          | **0.019**    | **1.014**         | **0.335**       |
> | 2K      | 0.026              | 0.023        | 1.017             | 0.338           |
> | 4K      | 0.030              | 0.029        | 1.020             | 0.340           |
> | 8K      | 0.036              | 0.036        | 1.025             | 0.347           |
> | 16K     | 0.041              | 0.042        | 1.036             | 0.416           |
>
> We find that **longer-context models exhibit higher loss volatility, less smooth loss curves, more frequent upward loss spikes, and larger gradient norms**, indicating less stable optimization dynamics during training. This suggests that **short-context pretraining is inherently more stable**, potentially providing more reliable gradient updates and facilitating better convergence.
>
> Due to the rebuttal policy, we are unable to present figures to visualize the dynamics. We will plot these interesting results in the updated version of our paper.
>
> ---
>
> ***Why does the proposed SkyLadder method fail to improve performance on certain reading comprehension tasks, as shown in Table 2?***
>
> We note that in the reading comprehension setup, Random+SkyLadder performs the best across all tasks. In contrast, for the IntraDoc setup, adding SkyLadder is overall comparable to the baseline. For tasks improved by SkyLadder, the difference is large. For tasks that with a performance drop, the decrease is very small (<0.8%). Our evaluation setup mainly follows [3], where a performance fluctuation is also observed at around 1-2%.
>
> Several factors may explain this fluctuation:
>
> - **Few-shot prompting noise**: We prepend 2-4 demonstrations with passages and questions to each test sample. The context thus includes extra noisy information that the model needs to handle, causing performance variations.
> - **Open-ended generation**: These QA tasks require the model to output free-form text, which naturally gives more variations than multiple-choice questions.
> - **Strict exact-match metric**: The generated text is evaluated by an exact match to the answer. Small differences in phrasing or format may be marked incorrect.
>
> All these factors may contribute to performance fluctuation. Even though  IntraDoc+SkyLadder does not seem to improve reading comprehension significantly, it still improves training efficiency and standard benchmark scores.
>
> Overall, we hope that we have addressed your concerns by carrying out the above analysis. We plan to include these results in the revised version of the paper, and would appreciate it if you could consider updating your score.
>
> ----
>
> ***References***
>
> [1] Barbero et al. Why do LLMs attend to the first token?. Arxiv 2504.02732
>
> [2] Kimi Team. Kimi K2: Open Agentic Intelligence. Technical Report 2025.
>
> [3] Zhao et al. Analysing The Impact of Sequence Composition on Language Model Pre-Training. ACL 2024

---

> > ### Comment · Reviewer_78Fj · 2025-08-05
> > **Reviewer Response After Rebuttal**
> >
> > Thanks to the authors for the thorough analysis, which solves my concerns. I remain positive about this paper, and please include these discussions in the next version.

---

> > > ### Author Response · Authors · 2025-08-05
> > >
> > > Thank you for your support and for confirming that our analysis addressed your concerns. We will make sure to incorporate these results into the next version of the paper.

---

### Official Review · Reviewer_FJPN · 2025-06-27

**Clarity:** 3
**Significance:** 3
**Originality:** 2
**Rating:** 4
**Confidence:** 2

**Summary:**

This paper reveals that models pretrained with shorter context windows consistently outperform their long-context counterparts under a fixed token budget. Motivated by this observation, the paper aims to better balance long-context capability with pretraining efficiency. To this end, it proposes SkyLadder, a method that progressively expands the context window during pretraining. SkyLadder can be integrated with strategies such as Random, BM25, and IntraDoc. Experimental results demonstrate that SkyLadder effectively achieves a trade-off between efficiency and long-context performance.

**Questions:**

See weaknesses

**Ethical Concerns:**

["NO or VERY MINOR ethics concerns only"]

**Final Justification:**

Most of my questions have been answered, but the only remaining question is whether this article can be generalised to larger LLMs. Since I am not an expert in this field, I cannot be sure that my assessment is entirely accurate.

**Limitations:**

yes

**Quality:**

3

**Strengths And Weaknesses:**

Strengths:
1. Interesting finding: models pretrained with shorter context windows consistently outperform their long-context counterparts under a fixed token budget.
2. The method is simple, effective, and efficient.
3. Experimental results and analysis are good.


 Weaknesses:
1. Is the 100B token budget used in Figure 1 sufficient? The observed results may be due to an insufficient number of tokens. I suggest conducting additional experiments with a larger token budget to validate the findings. In addition, is this due to the length characteristics of these nine datasets?

2. The experiments in this paper are conducted on models with up to 3B parameters. It remains unclear whether the conclusions can generalize to larger-scale language models.

3. It would be better to extend Tables 1 and 2 to include more pre-training corpora, rather than just the 100B CC.

4. It would be beneficial to evaluate the proposed method on a broader range of long-text benchmarks to provide a more comprehensive assessment.

5. Why does Random+SkyLadder outperform IntraDoc+SkyLadder in both Table 1 and Table 2? Is this due to the small context window?

6. Since the evaluation in this paper is highly related to input length, I suggest including information about the dataset lengths to provide a clearer presentation.

---

> ### Author Rebuttal · Authors · 2025-07-31
>
> Thanks for your valuable feedback. We are delighted that you find our work interesting. Here are our responses to each question:
>
> ***1. More tokens for Figure 1.***
>
> We conducted additional experiments by training 1B models from scratch on 200B Fineweb-Pro tokens, with a context length of 1K, 4K, and 16K. Results on downstream tasks are shown below:
>
> | Training Length | Avg   | ARC-E  | ARC-C  | CSQA  | HellaSwag | OBQA   | PIQA   | SocialIQA | Winogrande | MMLU   |
> |-----------------|-------|--------|--------|-------|-----------|--------|--------|-----------|------------|--------|
> | 1K              | **56.6** | **74.1** | **43.5** | **61.8** | 61.0     | **49.4** | **73.6** | **51.2** | **59.8**      | **34.6** |
> | 4K              | 54.5   | 71.7   | 40.6   | 58.1   | **61.1** | 45.2   | 71.3   | 49.7      | 58.1       | **34.6** |
> | 16K             | 53.7   | 69.1   | 40.5   | 56.8   | 60.9      | 43.2   | 72.6   | 48.6      | 57.9       | 33.9   |
>
> The trend remains consistent: shorter context windows lead to better overall performance under a fixed token budget. We plan to extend training to even larger budgets (e.g., 500B tokens) in future work.
>
> The following table represents the length characteristics of these datasets:
> | Metric | ARC-C  | ARC-E  | CSQA  | HellaSwag | OBQA  | PIQA  | SocialIQA | Winogrande | MMLU  |
> |--------|--------|--------|-------|-----------|-------|-------|-----------|------------|-------|
> | Mean   | 222    | 216    | 146   | 508       | 129   | 224   | 226       | 149        | 540   |
> | Std    | 20     | 16     | 7     | 32        | 9     | 29    | 6         | 4          | 512   |
> | Min    | 191    | 194    | 134   | 435       | 116   | 191   | 210       | 140        | 155   |
> | Max    | 401    | 336    | 203   | 576       | 192   | 440   | 262       | 170        | 3144  |
>
> These benchmarks are commonly reported when a base model is released. While these benchmarks are relatively short, they are standard for assessing a base model’s knowledge and reasoning abilities: we do not want a long-context model to hallucinate simply because the user provides a short query.
>
> ---
>
> ***2. Generalization to larger models.***
>
> We also hope to validate SkyLadder on larger models like 7B. However, such experiments require more tokens and compute, which is infeasible during the rebuttal period. Our current results already demonstrate generalizability across multiple model sizes, context lengths, datasets, and tasks. As shown in Table 5, the relative gains of SkyLadder increase with model size (from +1.1% for 120M to +3.5% for 3B), indicating that the benefits are likely to be even larger as model size grows.
>
> ---
>
>
> ***3. Extend Table 1 and 2 to more corpora.***
>
> We reported the performance of pretraining on Fineweb-Pro in Table 4. We report the performance of these models on standard benchmarks below:
>
>
> | Model      | Average | ARC-E | ARC-C | CSQA | HS   | OBQA | PIQA | SIQA | WG   | MMLU |
> |------------|---------|-------|-------|------|------|------|------|------|------|------|
> | Random     | 52.5    | 69.0  | 39.2  | 56.8 | 55.7 | 43.0 | 70.2 | 48.5 | 56.0 | 33.7 |
> | + SkyLadder | **55.2** | 72.6  | 40.3  | **60.7** | **60.7** | **48.4** | 71.8 | **49.1** | **58.1** | **35.3** |
> | IntraDoc   | 54.3    | 72.2  | 41.0  | 58.3 | 59.8 | 45.4 | **72.8** | 48.7 | 55.8 | 34.7 |
> | + SkyLadder  | 54.8    | **72.8** | **42.9** | 58.2 | 59.6 | 46.2 | 72.4 | 48.6 | 57.6 | **34.8** |
>
> Models with SkyLadder added show steady performance gains on these commonly used benchmarks, even when trained on a different corpus.
>
> For benchmarks listed in Table 2, the scores are as follows:
>
> | Model      | RC Avg. | HotpotQA | SQuAD | NQ   | TriviaQA | RACE-h | Long Avg. | MDQA | RULER |
> |------------|---------|----------|-------|------|-----------|--------|-----------|------|--------|
> | Random     | 20.4    | 1.9      | **25.4**  | 15.0 | 25.4      | **34.0**   | 11.1      | 15.5 | 6.7    |
> | + SkyLadder  | **21.2**    | **7.5**      | 24.3  | 13.9 | **26.5**      | 33.8   | 12.2      | 15.9 | 8.4    |
> | IntraDoc   | 20.2    | 5.9      | 24.1  | 11.8 | 25.3      | 33.7   | 12.1      | 14.8 | 9.5    |
> | + SkyLadder  | 20.7    | 6.6      | 23.3  | **14.0** | 25.8      | 33.9   | **13.9**      | **17.3** | **10.4**   |
>
> SkyLadder achieves better or comparable performance on these benchmarks as well, demonstrating its generalizability.
>
> ---
>
>
>
> ***4. More long-context benchmarks.***
>
> We evaluate our largest 3B model on additional long-context tasks (Tables 12 and 13 in Appendix A.7.3). Key results are summarized below:
>
> First, we follow a related work [1] to evaluate models on TOEFL and QuALITY. SkyLadder achieves better performance on these benchmarks.
>
> | Model      | TOEFL | QuALITY |
> |------------|-------|---------|
> | Baseline   | 43.5  | 30.6    |
> | + SkyLadder  | **48.0**  | **30.9**    |
>
>
> **Many-shot ICL**: We follow [2] and [3] to test on text classification. Numbers in brackets are the number of labels for each task.
> | Model      | DBpedia (14) | AGNews (4) | Amazon (2) | Yelp (2) | SST2 (2) | Average |
> |------------|--------------|------------|------------|----------|----------|---------|
> | Baseline   | 17.4         | 68.6       | **94.3**   | 94.7     | **94.5** | 73.9    |
> | + SkyLadder  | **25.5**     | **75.8**   | 94.1       | **95.0** | 92.2     | **76.5**|
>
>
> SkyLadder shows significant gain for tasks with many labels. Both models achieve comparable performance on binary tasks, possibly because the performance is almost saturated above 90%.
>
> **Retrieval-Augmented Generation**: We follow [4] for evaluating long-context RAG performance on multiple QA datasets. We fill the entire 8K context for this setup.
>
> | Model      | NQ   | TriviaQA | HotpotQA | PopQA | Average |
> |------------|------|----------|----------|-------|---------|
> | Baseline   | 24.3 | 45.2     | 29.3     | 22.5  | 30.3    |
> | + SkyLadder  | **27.8** | **52.7**    | **32.3**    | **29.3**  | **35.5**    |
>
> SkyLadder outperforms the baseline by a large margin across all QA datasets in the RAG task, which is an important application scenario for long context models.
>
> Overall, as shown by these additional results, SkyLadder can be better than or comparable with the baseline on multiple long context evaluations while achieving significant performance gains on standard benchmarks.
>
> ---
>
>
> ***5. Random + SkyLadder outperforms IntraDoc + SkyLadder***
>
> We thank the reviewer for spotting this intriguing phenomenon. First, we note that both SkyLadder and IntraDoc create shorter contexts; thus, the gain of applying SkyLadder to IntraDoc is smaller compared to Random. Towards the end of training, Random+SkyLadder (denoted as RS) is trained on overall longer contexts compared to IntraDoc+SkyLadder (IS): the RS model is able to see contexts beyond the current document. Thus, RS is exposed to more diverse and noisy multi-document contexts. Such exposure likely teaches the model to filter out irrelevant content and focus on key information. Thus, it shows better performance on the benchmarks, especially the longer ones with noisy contexts.
>
> ---
>
>
> ***6. Information about dataset lengths***
>
> As we are unable to attach figures in the rebuttal, we have to present the statistics as a table. We first show the distribution of the pretraining corpora we used:
>
> | Dataset     | Mean | Median | StdDev | Min | Max   | P25  | P75  | P90  | P95  | P99   | Skewness | Kurtosis |
> |-------------|------|--------|--------|-----|--------|------|------|------|------|--------|------|------|
> | CommonCrawl | 1973 | 1067   | 4567   | 45  | 594272 | 651  | 1867 | 3601 | 6005 | 16261 | 21   | 820  |
> | Fineweb Pro | 1364 | 849    | 2295   | 1   | 230949 | 507  | 1481 | 2557 | 3810 | 9733  | 15   | 533  |
>
> Both distributions are right-skewed, which means long documents are rare. Overall, the CC corpus has longer documents, as Fineweb-Pro has been carefully cleaned and processed.
>
> As for the evaluation datasets, besides the ones we reported earlier in \#1 in this response, the statistics for other benchmarks are as follows:
>
> | Metric | MDQA | RULER | SQuAD | HotpotQA | NQ  | TriviaQA | RACE |
> |--------|------|-------|-------|----------|-----|----------|------|
> | Mean   | 5150 | 7259  | 1048  | 5010     | 583 | 566      | 492  |
> | Std    | 287  | 745   | 81    | 993      | 21  | 28       | 121  |
> | Min    | 4172 | 6209  | 923   | 3587     | 536 | 529      | 122  |
> | Max    | 6755 | 8061  | 1174  | 7842     | 633 | 643      | 1323 |
>
> We will put more dataset information in the revised version of the paper.
>
> We hope the above response addresses your concerns, and we would appreciate it if you could consider raising your score.
>
> ---
>
> ***References***
>
> [1] Pouransari et al. Faster LLM Training with Variable Sequence Length Curriculum. NeurIPS 2024.
>
> [2] Zhao et al. Analysing the Impact of Sequence Composition on Language Model Pre-training. ACL 2024.
>
> [3] Shi et al. In-context Pretraining: Language Modeling Beyond Document Boundaries. ICLR 2023.
>
> [4] Yen et al. HELMET: How to Evaluate Long-context Language Models Effectively and Thoroughly. ICLR 2025.

---

> > ### Comment · Reviewer_FJPN · 2025-08-02
> >
> > Thank you for your reply. Some of my questions have been resolved. However, as I am not an expert in this field, I cannot be certain that my assessment is entirely accurate. Therefore, I have maintained my positive score and continue to support the acceptance of this paper.

---

> > > ### Author Response · Authors · 2025-08-04
> > >
> > > Thank you for your helpful review and for maintaining a positive view of our work. We truly appreciate your time and support.

---

### Official Review · Reviewer_Tqtf · 2025-06-29

**Clarity:** 3
**Significance:** 2
**Originality:** 2
**Rating:** 4
**Confidence:** 5

**Summary:**

This paper proposes SkyLadder, a simple yet effective context window scheduling strategy for pretraining LLM. Instead of using a fixed long context window throughout training, SkyLadder gradually increases the window size from small to large, improving both model performance and training efficiency. Through extensive experiments on models up to 3B parameters and context lengths up to 32K, the authors show that SkyLadder consistently outperforms fixed-window baselines across standard benchmarks, long-context tasks, and multiple masking strategies, while also reducing computational cost.

**Questions:**

1. What training scale was used in the "Continual Pretraining" experiment (Tab. 7)? Was an evenly spaced expansion (e.g., 4K → 8K → 16K → 32K) used for context window sizes?
2. Does SkyLadder affect the model's ability to generalize to unseen sequence lengths? Is there a risk that the model’s generalization ability to longer sequences may be compromised by the gradual expansion of the context window during training?

**Ethical Concerns:**

["NO or VERY MINOR ethics concerns only"]

**Final Justification:**

The response addresses W1 to some extent; however, I am not fully satisfied, as attention masking appears to be a general design choice. W2 and W3 remain unaddressed, and the authors should demonstrate the applicability of the method. The other questions have been addressed appropriately. After considering the other reviews, I will not oppose accepting this paper; however, I will not advocate for its acceptance either.

**Limitations:**

Yes

**Quality:**

3

**Strengths And Weaknesses:**

Strengths
1. The paper conducts extensive and sufficient experiments to support its conclusions.
2. The proposed context window scheduling strategy offers valuable insights for training long-context LLM
3. The proposed SkyLadder method improves both model performance and training efficiency.

Major Weaknesses
1. The paper provides solid experimental evidence for the effectiveness of context window scheduling, but the overall innovation is limited.
2. Although the authors note that experiments were limited to models below 3 billion parameters due to computational constraints, the absence of validation on larger-scale models (particularly 7B-level LLM), undermines the persuasiveness of the method's scalability and practical applicability.
3. Although the paper conducts a series of experiments based on the TinyLlama baseline, the lack of comparative analysis with other advanced models, such as LLaMA 3.2 3B, Mistral 3B, and Qwen 2.5, makes it difficult to demonstrate the superiority of the proposed method.

---

> ### Author Rebuttal · Authors · 2025-07-31
>
> Thanks for your valuable feedback. We appreciate that you find our study insightful. Here are our responses to your concerns in the Weaknesses section:
>
> ***The paper provides solid experimental evidence for the effectiveness of context window scheduling, but the overall innovation is limited.***
>
> Thanks for acknowledging the rigor of our study. We want to clarify that the main contribution of our paper is to provide a new insight that context window scheduling can boost the downstream task performance, which is "novel" compared to previous works. As discussed in the Related Work section, previous works that explore related ideas are mostly designed to accelerate training, without demonstrating consistent performance gains. Our work is the first to conduct a large-scale, controlled study (up to 3B models, 32K context) showing that context window scheduling not only improves training efficiency but also significantly boosts downstream task performance.
>
> Furthermore, unlike approaches that select documents of certain lengths (which risk introducing domain biases), SkyLadder dynamically adjusts the effective context window through attention masking, mitigating the distribution shift issues in long-context training. This combination of scale, performance improvements, and bias mitigation constitutes an important contribution beyond prior works.
>
>
> ---
>
> ***Scaling to larger models.***
>
> We are also curious about further scaling and acknowledge this as a limitation. We are unable to finish the pretraining of 7B models during the rebuttal period, as training a larger model naturally requires more tokens beyond our compute capability. However, we believe our findings have provided strong evidence of scalability:
>
> - **Positive scaling trend**: In Table 5, As shown in Table 5, the performance gain from SkyLadder grows larger as model size increases (from 120M to 3B parameters), suggesting its potential in larger models.
> - **Consistent gain across different settings**: We demonstrate consistent improvements across different corpora, masking strategies, evaluation tasks, and model architectures. This shows that SkyLadder is not specific to a single configuration and should generalize to larger models
>
> These results support the scalability of our method to larger models, which we leave as future work.
>
> ---
>
> ***The lack of comparative analysis with other advanced models, such as LLaMA 3.2 3B, Mistral 3B, and Qwen 2.5.***
>
> We appreciate the reviewer's suggestion to include comparisons with advanced models. However, a direct comparison is not meaningful due to fundamental differences in training setups:
> - First, our paper looks at pretraining a model *from scratch*. These public LLMs are trained on orders of magnitude more data (>10T tokens), and their superior performance is largely attributed to the large data scale.
> - The pretraining data for these models is not publicly available, making it impossible to control for differences in data quality or preprocessing.
> - Our goal is not to release a new state-of-the-art model, but to isolate and study the effect of context scheduling under a fixed token budget, where we can rigorously control for all other variables.
>
> Our 3B model exactly follows the Llama3.2 3B architecture (Table 10), and we train two models on exactly the same tokens to verify the effectiveness. This is a strictly controlled setup that rules out confounding factors such as improved data cleaning.
>
> For other architectures, we note that Mistral has no 3B model officially. Therefore, we additionally pretrain Qwen 0.5B models from scratch on 100B tokens to show the generalizability of our approach. The table below shows the PPL on the held-out validation set.
>
> | Model      | PPL (1K) | PPL (4K) | PPL (8K) |
> |------------|----------|----------|----------|
> | Random   | 14.8     | 13.1     | 12.5     |
> | + SkyLadder  | **14.3**     | **12.7**     | **12.1**     |
>
> SkyLadder improves the validation PPL across all evaluation lengths.
>
> | Model      | Average | ARC-E | ARC-C | HellaSwag | MMLU  | CSQA  | OBQA  | SocialIQA | PIQA  | WinoGrande |
> |------------|---------|-------|--------|-----------|-------|--------|--------|-------------|--------|--------------|
> | Random   | 42.6    | 51.9  | 28.5   | 39.3      | 28.6  | 43.7   | 33.2   | **43.9**   | 61.2  | **53.4**     |
> | + SkyLadder  | **44.3**| **55.6**| **31.0** | **40.3**  | **29.7**| **48.2**| **35.2**| 43.7       | **62.4**| 52.2         |
>
> On common benchmarks, SkyLadder also performs better overall. These results confirm that our method is applicable to Qwen models and generalizes across architectures. We have also verified its effectiveness on hybrid-attention models such as Gemma 3 (Table 17; Appendix Line 742).
>
> ---
> Here are our responses to the specific questions in the “Questions” section.
>
> ***What training scale was used in the "Continual Pretraining" experiment (Tab. 7)? Was an evenly spaced expansion (e.g., 4K → 8K → 16K → 32K) used for context window sizes?***
>
> Table 7 shows results for 1B models pretrained on 100B high-quality Fineweb-Pro tokens, with a final target length of 32K. We did not employ gradual expansion for the Cont. Pretrain setup in Table 7. However, we agree that it is interesting to examine the evenly spaced expansion as suggested, and we conducted additional experiments. The results are shown below:
>
> | Model                                 | Standard Avg. | Long Avg. | MDQA | RULER |
> |--------------------------------------|----------------|-----------|------|-------|
> | Constant                             | 50.7           | 9.7       | 12.8 | 6.6   |
> | SkyLadder                            | **54.3**       | **13.5**  | **18.4** | **8.5**   |
> | Cont Pretrain (4K -> 8K -> 16K -> 32K) | 52.9           | 10.4      | 15.3 | 5.4   |
>
>
> Although this gradual continual pretraining setup outperforms the baseline, SkyLadder still achieves the best performance across all tasks. This result indicates that the schedule should be applied from the beginning of pretraining, rather than only during the continual pretraining stage.
>
> ---
> ***Does SkyLadder affect the model's ability to generalize to unseen sequence lengths?***
>
> Thanks for this interesting question. Our answer is that it **improves the generalization ability to unseen lengths**, as shown by the following observations.
>
> First, we directly evaluate pretrained checkpoints trained with an 8K window on sequences longer than 8K, following the protocol in [1]. The table below shows the perplexity (PPL) on the GovReport dataset for different evaluation lengths (each column) up to 32K:
>
> | Model      | 8K   | 12K   | 16K   | 24K   | 32K   |
> |------------|------|-------|-------|-------|-------|
> | Random     | 6.9  | 24.4  | 99.4  | 364.1 | 579.8 |
> | + SkyLadder  | **6.7**  | **9.6**   | **22.0**  | **62.6**  | **99.5**  |
>
> Interestingly, the SkyLadder model, with gradual expansion, achieves lower perplexity beyond the training length. In contrast, the PPL of the Random baseline quickly explodes.
>
> Second, we also implemented long-context continual pretraining, where both models are extended from 8K to 32K using 2B additional tokens from [2]. We evaluate the models on the RULER benchmark.
>
> | Model      | 32K  | 16K  | 8K   | 4K   |
> |------------|------|------|------|------|
> | Random     | 27.2 | 38.6 | 41.8 | 51.1 |
> | + SkyLadder  | **29.8** | **40.3** | **47.9** | **55.7** |
>
> In this case, the SkyLadder model outperforms the baseline as well. This confirms that gradual context expansion leads to a stronger model with better generalization to unseen lengths.
>
> We hope our responses have addressed your concerns and clarified the contributions of our work. We would greatly appreciate it if you could consider raising your score.
>
> ---
> ***References***
>
> [1] Peng et al. YaRN: Efficient Context Window Extension of Large Language Models. ICLR 2024.
>
> [2] Fu et al. Data Engineering for Scaling Language Models to 128K Context. ICML 2024.

---

> > ### Comment · Reviewer_Tqtf · 2025-08-05
> > **Official Comment by Reviewer Tqtf**
> >
> > I appreciate all the detailed explains to my concerns. After carefully referring to  all other reviews, I decide to raise my rating.

---

> > > ### Author Response · Authors · 2025-08-05
> > >
> > > Thank you for raising your score after considering our rebuttal. We appreciate your support and the insightful questions that help us improve the paper.

---

### Note · Authors · 2025-08-12

We appreciate the positive view shared by all reviewers for the high quality and solid experiments in our paper. Three reviewers responded that our responses addressed their concerns. After the rebuttal, Reviewer Tqtf decided to raise the score, and Reviewer 78Fj and FJPN mentioned that they “remain positive” and “support the acceptance” of this paper.


Below, we summarize the key discussion points during the rebuttal:
- **Contribution of our work**:
    - We conduct controlled experiments on the effect of context window on pretraining. We obtain a counterintuitive finding that naive long-context training is not optimal for either long or short tasks, challenging the prevailing practice of pretraining with ever-longer contexts.
    - Beyond previous works that mainly focused on training speedup, we demonstrate that context window scheduling also greatly enhances model performance.
    - We systematically investigate the effect of scheduling and find substantial performance variation among different schedule types. This suggests that context length is an important dimension for scheduling and deserves more open research.
    - SkyLadder controls the context length by applying attention masks, rather than selecting documents of certain lengths. This avoids domain bias and integrates cleanly with any packing strategies.
- **Generalizability of SkyLadder**: We demonstrate the effectiveness of our approach across different model sizes, context lengths, and pretraining corpora. In our responses to Reviewer Tqtf and hz6k, we also verify that it generalizes better to unseen sequence lengths and continues to outperform during continual pretraining.
- **Analysis of training dynamics**: We find that SkyLadder exhibits more effective attention patterns and stabilizes training. It delays the emergence of an attention sink by training on short contexts, and gives smoother loss curves with fewer spikes (in response to Reviewer 78Fj).


Overall, we thank all reviewers for the thoughtful comments and will incorporate these clarifications and new results into the final paper.

---

### Decision · Program_Chairs · 2025-09-17

**Decision:**

Accept (poster)

**Comment:**

The paper analyzes the effect of context window length on LLM pretraining and, based on these observations, introduces SkyLadder, a short-to-long context window curriculum. The method improves accuracy under a fixed token budget on standard tasks, while matching or exceeding performance on long-context tasks, with similar training speed improvements as well.

While some observations (e.g., the effect of context length during training) are already known in the community, and the method itself is not highly novel (e.g., related to long-context fine-tuning), the reviewers agreed that the context length analysis is well-designed and the experiments are thorough, spanning datasets, model sizes, and hyperparameters. The motivation is clear, and the proposed method is simple and effective. A remaining concern is that experiments extend only up to 3B parameters; the rebuttal offered some arguments, based on scaling trends, that the method should generalize to larger models.

Overall, the AC agrees that the paper makes a solid empirical contribution, with carefully designed analysis and a simple method that could be of significant practical benefit to the community. The paper is therefore recommended for acceptance.